# Socio-hydrological spaces in the Jamuna River floodplain in Bangladesh

Md Ruknul Ferdous[1, 2], Anna Wesselink[1], Luigia Brandimarte[3], Kymo Slager[4], Margreet Zwarteveen[1, 2], Giuliano Di. Baldassarre[1, 5, 6]

[1]Department of Integrated Water Systems and Governance, IHE Delft Institute for Water Education, 2611 AX, Delft, The Netherlands
[2]Faculty of Social and Behavioural Sciences, University of Amsterdam, 1012 WX, Amsterdam, The Netherlands
[3]Sustainability Assessment and Management, KTH, SE-100 44, Stockholm, Sweden
[4]Deltares, 2600 MH, Delft, The Netherlands
[5]Department of Earth Sciences, Uppsala University, SE-75236 Uppsala, Sweden
[6]Centre of Natural Hazards and Disaster Science, CNDS, SE-75236 Uppsala, Sweden

*Correspondence to*: Md Ruknul Ferdous (r.ferdous@un-ihe.org)

**Abstract.** Socio-hydrology aims to understand the dynamics and co-evolution of coupled human-water systems, with research consisting of generic models as well as specific case studies. In this paper, we propose a concept to help bridge the gap between these two types of socio-hydrological studies: socio-hydrological spaces (SHS). A socio-hydrological space is a geographical area in a landscape. Its particular combination of hydrological and social features gives rise to the emergence of distinct interactions and dynamics (patterns) between society and water. Socio-hydrological research on human-flood interactions has found two generic responses, 'fight' or 'adapt'. Distilling the patterns resulting from these responses in case studies provides a promising way to relate contextual specificities to the generic patterns described by conceptual models. Through the use of SHS, different cases can be compared globally without aspiring to capturing them in a formal model. We illustrate the use of SHS for the Jamuna floodplain, Bangladesh. We use narratives and experiences of local experts and inhabitants to empirically describe and delimit SHS. We corroborated the resulting classification through the statistical analysis of primary data collected for the purpose (household surveys and focus group discussions) and secondary data (statistics, maps etc.). Our example of the use of SHS shows that the concept draws attention to how historical patterns in the co-evolution of social behaviour, natural processes and technological interventions give rise to different landscapes, different styles of living, and different ways of organizing livelihoods. This provides a texture to the more generic patterns generated by socio-hydrological model, promising to make the resulting analysis more directly useful for decision makers. We propose that the usefulness of this concept in other floodplains, and for other socio-hydrological systems than floodplains, should be explored.

**1 Introduction**

The hydrological sciences community has recently launched socio-hydrology as one of the research themes of the current scientific decade of the International Association of Hydrological Sciences (IAHS) "Panta Rhei – Everything Flows" (2013-2022), which aspires to 'advance the science of hydrology for the benefit of society' (Montanari et al., 2013 p.1257). Socio-
hydrology aims to understand the dynamics and co-evolution of coupled human-water systems (Sivapalan et al., 2012). In traditional hydrology, humans are either conceptualised as an external force to the system under study, or taken into account as boundary conditions (Milly et al., 2008; Peel and Bloschl, 2011). In socio-hydrology, human factors are considered an integral part of the system. Understanding such coupled system dynamics is expected to be of high interest to governments who are dealing with strategic and long term water management and governance decisions (Sivapalan et al., 2012).

As in any newly defined research area, socio-hydrology researchers are looking to determine how to implement their shared goal. This has resulted in a number of overview or position papers (e.g. Blair & Buytaert, 2016; Sivapalan, 2015; Pande & Sivapalan, 2017; Sivapalan & Bloschl, 2015; Troy et al. 2015) as well as several case studies (e.g. Gober & Wheater, 2014; Kandasamy et al., 2014; Liu et al., 2014; Mehta et al., 2014; Srinivasan, 2015; Mostert, 2018). In the discussions, the use of conceptual and deterministic models to analyse concrete situations is an important topic. As in any attempt to produce
insights that transcend specific cases, methods of abstraction from reality to find causal relationships and stylised equations (generalisation) are sometimes difficult to reconcile with more detailed representations of what is happening in a specific location (Blair & Buytaert, 2016). While enabling global comparison by using data sets from different locations, generic models unavoidably foreground some elements or dimensions of flood-society dynamics to the neglect of others (Magliocca et al., 2018). On the other hand, attempts to generalize from case-specific detailed models need to be looked at critically in
terms of the comparability and commensurability of the modelled phenomena with what happens elsewhere: detailed causal relationships in one case do not usually correspond to those in other cases (e.g. Elshafei et al. 2014).

In this paper, we focus on another, less formal way to capture socio-hydrological dynamics than causal relationships and models: patterns. Pattern detection is no new activity in socio-hydrology, because patterns are at the basis of the stylised representations (equations) in generic models. Historical patterns are foundational for a full understanding of generic as well
as place-based models, and pattern finding reinforces the feedback between empirical studies and modelling studies. However, we propose a new socio-hydrological concept to operationalise the search for patterns in the messy reality of specific cases: socio-hydrological spaces (SHS). Eventually, patterns found in cases may be formalised into causal relationships, but this does not necessary have to be the goal. We contend that patterns in and by themselves are valuable research results, especially in policy development and where data are scarce (Section 8). The SHS concept will be defined
and its implementation explained in Section 3. To illustrate how the concept can be used, we analyse human-flood interactions in the Jamuna floodplain, Bangladesh, making use of the two generic responses to flood risk 'fight' or 'adapt' that were found in earlier research on human-flood interactions (Section 2). In the Jamuna floodplain the differences between land and water are temporary and shifting, as is the size of the human population. The application of SHS allows capturing

the different socio-hydrological patterns that result from different societal choices on how to deal with rivers, floods and erosion, which in turn produce different living conditions and watery environments (Sections 4 and 5).

The detection of patterns in socio-hydrological relationships can be based on the interpretation of a combination of qualitative and quantitative data; it is therefore more feasible where quantitative data are scarce. This mid-level theorizing on the basis of empirically observable patterns was identified by Castree et al. (2014) as a desirable way forward in environmental research, as it makes it easier to link and translate model-deduced patterns with experienced realities. By providing locally relevant details and texture to more generically deduced patterns, SHS provides a useful methodological addition to the socio-hydrological understanding of floodplains. Its usefulness to other contexts such as irrigated catchments or urban water systems could also be investigated.

## 2 Patterns in the socio-hydrology of floodplains: fight or adapt

One type of situation that is relatively well studied by socio-hydrologists is the co-evolution of human societies and water in floodplains. After all, the existence of interdependencies between societies and their natural environment is particularly obvious in floodplains. Since the beginning of human civilization, many societies have developed in floodplains along major rivers (Vis et al., 2003). In spite of periodical inundations, a distinct preference for floodplain areas as places to settle and live in stems from their favourable conditions for agricultural production and transportation, enabling trade and economic growth (Di Baldassarre et al., 2010). Yet, floodplain societies have to learn how to deal and live with periodic floods and the relocation of river channels by erosion and deposition (Sarker et al., 2003). In general terms, floodplain societies do this by evaluating the costs of flooding and erosion against the benefits that rivers bring, and deciding whether to try to mitigate the risks by defending themselves against floods ('fight'), or to live with floods ('adapt'), or any combination of the two (Di Baldassarre, et al., 2013a,b). Whether and how societies can fight or adapt to flooding depends on the society's economic and technological possibilities. Therefore, 'fight' and 'adapt' are the two generic responses in the socio-hydrological dynamics of human-flood interaction. These combine differently in different contexts and locations, resulting in different socio-hydrological patterns.

For flood mitigation, societies have usually relied on engineering measures like embankments or levees to prevent flooding, and bank protection and spurs or guide bunds stretching into the river to prevent erosion. These measures can be seasonal (temporary) or permanent, and have more or less effect on flood prevention (Sultana et al., 2008). The construction of flood control measures might in turn alter the frequency and severity of floods, leading to a dynamic interaction between the river and the society living alongside it (Hofer and Messerli, 2006). An alternative response to flooding is adaptation. In order to adapt to flood risks, societies may limit costly investments in property or make them movable, adjust cropping patterns or choose crops that can cope with flooding, or move away altogether if alternative locations for settlement are available. Even when flood protection measures are in place, residual risks may necessitate adaptation measures. This means that in any real situation the two responses of 'fight' and 'adapt' are usually found together in a site-specific configuration, depending on

socio-economic, institutional, and natural conditions. We label the areas where the proportions are analogous due to similar conditions 'socio-hydrological spaces' (SHS) (see Section 3).

The study of floodplains using a socio-hydrological approach has advanced rapidly in the last few years (Di Baldassarre et al., 2013a,b; 2015; O'Connell and O'Donnell, 2014; Viglione et al., 2014, Chen et al., 2016; Grames et al., 2016; Ciullo et al., 2017; Barendrecht et al., 2017; Yu et al., 2017). In this research, the two responses to flooding 'fight' and 'adapt' take centre stage. The overall aim of this work is to further understanding on 'how different sociotechnical approaches in floodplains are formed, adapted, and reformed through social, political, technical, and economic processes; how they require and/or entail a reordering of social relations leading to shifts in governance and creating new institutions, organizations, and knowledge; and how these societal shifts then impact floodplain hydrology and flooding patterns' (Di Baldassarre et al., 2014, p.137).

Two different methodologies for the study of floodplains can be broadly distinguished, in parallel with general trends in socio-hydrology found by Wesselink et al. (2017). The first approach presents a narrative representation of the floodplain's socio-hydrological system. The narrative is generally based on qualitative research, often informed by experiences and knowledges of local experts and inhabitants about histories of living with floods, but may also include statistical data e.g. on trends. The resulting studies describe historical patterns in the co-evolution of river dynamics, settlement patterns and technological choices (Di Baldassarre et al., 2013a, 2014). Not all researchers who engage in this kind of studies identify their work as belonging to socio-hydrology (e.g. Van Staveren and Tatenhove, 2016, Van Staveren et al., 2017a, 2017b). In this qualitative research, the actual societal choices between 'fight' and 'adapt' are descriptively represented, without formalisation.

The second approach to studying socio-hydrological dynamics of floodplains focusses on the development and use of a generic conceptual model of human-nature interactions, which is subsequently expressed in terms of differential equations (e.g. Di Baldassarre et al., 2013b, 2015; reviewed in Barendrecht et al., 2017). This second approach also starts with a narrative understanding of the situation, in which patterns are key for deriving causal relationships. These narratives narrow down complex realities to a selection of phenomena and elaborate trends and causal relationships that are subsequently captured in mathematical models (see Elshafei et al. 2014 for a clear example of the role of narratives). Generic models aim to explain the feedback mechanisms that produce certain phenomena (often paradoxes or unintended consequences) that have been observed in many places around the world. For example, the stylised models of human-flood interactions introduced by Di Baldassarre et al. (2013b) use a mathematical formalization of a fundamental hypothesis: the levee effect (White, 1945) is explained by a decrease in risk awareness when flooding becomes less frequent because of the introduction (or reinforcement) of structural protection measures. This generic model has been used to explore and compare alternative scenarios of floodplain development (Di Baldassarre et al., 2015; Viglione et al., 2014). Current research includes further refinement (Grames et al. 2016; Yu et al., 2017) or comparison of this generic model to actual data for specific cases (Ciullo et al., 2017; Di Baldassarre et al., 2017). Yet, as societal responses to hydrological changes (including flood occurrences)

are 'very complex and highly unpredictable as it strongly depends on economic interests and cultural values' (Di Baldassarre et al., 2015 p.4780), formalisation is challenging.

## 3 Socio-hydrological spaces defined

To reflect pattern detection as intermediary activity between modelling and reality, we define a socio-hydrological space in two ways: from the empirical observations, which may include quantitative data but also general contextual knowledge ('bottom-up'), and from the conceptual models of the general patterns found in human-flood interactions ('top-down'). Starting from the empirical observations captured in quantitative and qualitative data, we define a socio-hydrological space as a geographical area in the landscape with distinct hydrological and social features that give rise to the emergence of distinct interactions and dynamics between society and water. Starting from the generic patterns captured in conceptual modelling, a socio-hydrological space is the empirical expression of a specific combination of generic responses (here: fighting and adaptation dynamics) in a geographical area that is distinct from the neighbouring one. Importantly, both definitions apply simultaneously, and are operationalised in an iterative manner to study the socio-hydrology of an area as shown in the example for the Jamuna flood plain (Sections 4 to 6).

Using SHS in the analysis of socio-hydrological dynamics helps to make the necessary intermediary step between the messy and many details used to characterize a specific location (space) and the stylised abstraction of generic models. With the proposal of SHS we are looking for a middle ground where we preserve the variability of reality and the unpredictability of human behaviour and decisions, not force-fitting these into a model, while at the same time recognising patterns (due to combinations of similar or comparable fight and/or adapt responses). We thus propose that SHS can serve the function of a lens through which to view and filter the complex reality of specific cases, in order to find patterns in human-water interactions. Such patterns can then be compared and contrasted to patterns in other locations to see if further generalisation towards generic models is possible. The use of SHS invites the researcher to have an open mind to the existence of expected or unexpected patterns in the location under investigation, using a thorough understanding of the specifics of this location in terms of society, history, economics, natural system, technical interventions, etc.  Insights from one location can then be compared to analyses of other cases in order to explore whether the same or different patterns occur, and for the same reasons. These patterns can then be generalised through a more formal conceptualisation of socio-hydrological systems, whereby the existing conceptual models may be taken as a starting point. On the one hand SHS thereby relates to a specific space, on the other hand it helps to find general patterns of human-water interactions, which means that use of SHS to analyse different cases enables global comparison.

It is interesting to note that some of the earlier socio-hydrological research on floodplains can be said to implicitly employ something resembling the SHS concept (Fig. 1). In their study, which is partly based on the Po floodplain, Di Baldassarre et al. (2013a, 2014) identify two patterns of society-river interactions. In the 'adaptation effect' pattern the use of flood defence technology is limited, resulting in frequent flooding which is in turn associated with decreasing vulnerability (see also

Kreibich et al., 2017). The 'levee effect' pattern results when flood protection structures lead to less frequent but more severe flooding, which is in turn associated with increasing vulnerability (Di Baldassarre et al., 2015) (already identified by White, 1945; see also Kates et al., 2006). These two patterns can be rendered in terms of SHS, yielding a classification of:

a) the 'adaptation space' where frequent flooding results in less economic development and lower population density
5       and other human adjustments;

b) the 'fighting space' where flood protection structures lead to less frequent but more severe flooding, more economic development and higher population density and other human adjustments.

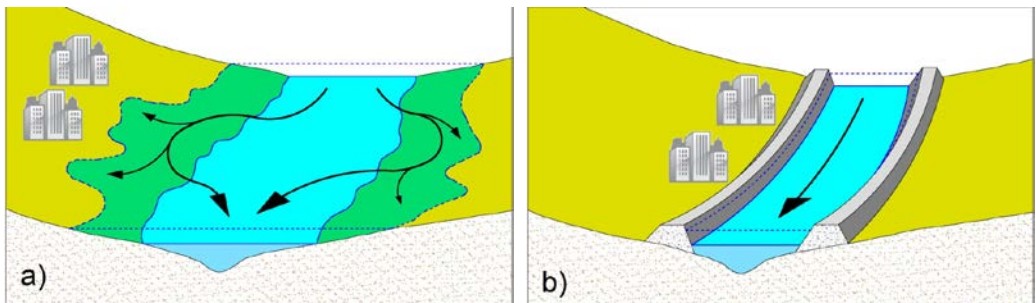

**Figure 1: Schematic of human adjustments to flooding: (a) adaptation: settling away from the river, and (b) fighting: raising levees**
**or dikes (after Di Baldassarre et al., 2013b).**

In these first conceptualizations, one floodplain is assumed to show one or the other pattern at one point in time, while allowing shifts over time from adaptation to levee effect (Di Baldassarre et al., 2013a). This classification categorises a floodplain as having one single socio-hydrological pattern ('fight' or 'adapt'). Di Baldassarre et al. (2015) then classify several floodplains worldwide in one of the two patterns. For example, they classify Bangladesh as a whole into the 'adapt'
type. However, it turns out that several sections of the floodplain in Bangladesh are protected by an embankment (see Section 5), with residual flood risks giving rise to adaptation behaviour. Similarly, their classification of the Rhine floodplain in The Netherlands as 'fighting floods' holds in general, though in several places adaptation is being experimented with (Wesselink et al. 2007, Van Staveren and Van Tatenhove, 2016). In the same country, the Meuse valley was classified as 'adaptation' although embankments have been added to protect built-up areas (Reuber et al., 2005; Wesselink et al., 2013).
As the goal of generic models is to describe decadal dynamics at large scale (Di Baldassarre et al., 2013b), they can only capture the main phenomena in large areas, such as a whole floodplain (in time or space) or a river basin. Instead, SHS induce the researcher to further refine the analysis of human-flood interactions from the generic to the more local where, for example, both response may co-exist at one time in specific proportions. In this way, SHS allow more specific and detailed representation of the reality of these interactions, while still enabling comparison between cases by referring to generic
patterns. In what follows, we illustrate how the concept can be used in a more detailed and refined analysis of the Jamuna floodplain in Bangladesh. We show how its use can provide nuances to the broad-sweep overall classification by showing that within this overall characterisation some areas to some extent exhibit a 'levee effect', while other areas do not fit the two-way classification.

To use the concept of SHS, we propose a two-step approach. First, the top down definition of SHS guides the researcher to look for the generic patterns in the information collected about the study area. As notes this information is based on a thorough understanding of a specific floodplain (geography, history, technology, societal occupation etc.). This results in a preliminary geographical delineation of distinct SHS and their qualitative descriptions by means of narratives, schematised drawings, maps etc.; these results have the function of being hypotheses in the next step. In the second step, quantitative data analysis is employed to confirm, reject or correct these initial hypotheses, that is, this analysis provides the data driven (bottom up) delineation of SHS. If the classification is not statistically significant, merging or splitting of categories should be considered as well as re-drawing the boundaries (repeat step 1). However, this adjustment should always be based on arguments based on a good understanding of the floodplain, since statistical significance by itself does not explain socio-hydrological dynamics.

Similar research methods were used before in socio-hydrology, e.g. geo-statistics to study the interaction between river bank erosion and land use (Hazarika et al., 2015), or so-called data-driven narratives (Treuer et al. 2017) and the pairing of statistical analysis and narratives (Hornberger et al., 2015, Mostert, 2018). While the combination of narrative and statistical methods that we use is therefore not new, their application to SHS enables transcending single case studies in the search for more generalizable patterns. We could therefore envisage that the methods used in Step 2 could be different, as long as they contribute to the goal of identifying and validating SHS.

The following case study demonstrates how the SHS approach can be used. Our goal is not to include all available data to provide an exhaustive analysis, but to show how SHS help to detect and understand socio-hydrological dynamics. The socio-hydrological characteristics and data availability guide the choice of methods in our socio-hydrological analysis of a part of the Jamuna floodplain in Bangladesh. In other circumstances the application of SHS will likely entail different variables and methods.

## 4 Research approach

### 4.1 Case study area

The delta where the Ganges, Brahmaputra and Meghna rivers meet the sea in the Bay of Bengal encompasses 230 river channels and covers most of Bangladesh (Mirza et al., 2003). It is the largest delta in the world draining almost all of the Himalayas, the most sediment-producing mountains in the world (Goodbred et al., 2003). The flows of the three rivers add up to an average of 1 trillion cubic meter per year of water and 1 billion tonnes per year of sediment. The sediment load is very high, resulting in very dynamic river channels (Allison, 1998). In the early 18th century, the main course of the current Jamuna was flowing through what is now the Old Brahmaputra, to the east of the Jamuna. Sometime between 1776 and 1830 the course of the Brahmaputra shifted from east to west, and the 'new' river was given the name Jamuna. Since then, the Jamuna has shown progressive westward migration and widening, meanwhile transforming from a meandering river to a braided one (CEGIS, 2007). The Brahmaputra Right Embankment (BRE) was constructed on the west bank of the Jamuna in

the 1960s to limit flooding and increase agricultural production, and also to try to stabilize the position of the river, the latter with limited success despite the addition of groynes and spurs.

Bangladesh is a very densely populated country with more than 140 million of people (964 persons per square km). Around 80% of the population lives in floodplain areas (Tingsanchali and Karim, 2005) and depends on agriculture and fisheries
(BBS, 2011). In the monsoon season, 25-30% of the floodplain area is inundated every year (Brammer, 2004). These 'normal' floods are valued by rural inhabitants because they are beneficial to the fertility of the land, provide ecosystem services (fish stock), and transportation (Huq, 2014). According to the classification by the Flood Forecasting and Warning Centre which categorises flooding events as normal, moderate and severe based on flood duration, exposure, depth and damage, extreme flood events were observed in 1954, 1955, 1974, 1987, 1988, 1998, 2004 and 2007 (FFWC/BWDB,
2017); the flood events since then were not judged extreme in the whole country, but in NW Bangladesh, which includes our study area, 2016 and 2017 were also extreme (FFWC/BWDB, 2017). Throughout the years, successive governments have implemented several flood control measures to protect agriculture and populations from floods (Sultana et al., 2008).

Riverbank erosion is associated with flooding in many areas of the country. The extremely poor people who live on the chars (islands in the big rivers) are most exposed to and affected by flood hazards and riverbank erosion. During the period 1973 to
2015, the net erosion was 90,413 ha and the net accretion 16,497 ha along the 220 km long Jamuna (CEGIS, 2016). Every year about 50,000 to 200,000 people are displaced by riverbank erosion, although they usually find another place to settle nearby in the area (Walsham, 2010). Hence, it is clear that hydrological processes (flooding and riverbank erosion) play a vital role in the way people in Bangladesh organize their lives, as manifested among others in patterns of migration, livelihoods and land use.
To understand these relationships between river and people better, this study focusses on a small area along approx. 30 km of the Jamuna River in the north of Bangladesh (Fig. 2). The total area is about 500 square km and the total population is approx. 0.36 million (BBS, 2011). The case study area includes parts of Gaibandha district and parts of Jamalpur district (Fig. 2). The total width of the case study area is around 24 km, of which the braided river bed takes approx. 12-16 km; this includes many inhabited river islands (chars) that flood with varying frequency (every year to only with severe floods). The
maintenance of the BRE in the study area has been sporadic. When constructed, the average height was 4.5m, width 6m and slope 1:3 on both sides (CEGIS, 2007). Though extreme discharges could not overtop this embankment, breaches have occurred which caused catastrophic floods and damages (RBIP, 2015). In the 2016 flood (observed during the field survey), the BRE was breached in Gaibandha district, resulting in a large area being flooded. On the left bank there is no human-made protection, but there is a natural levee that has been deposited by the river.

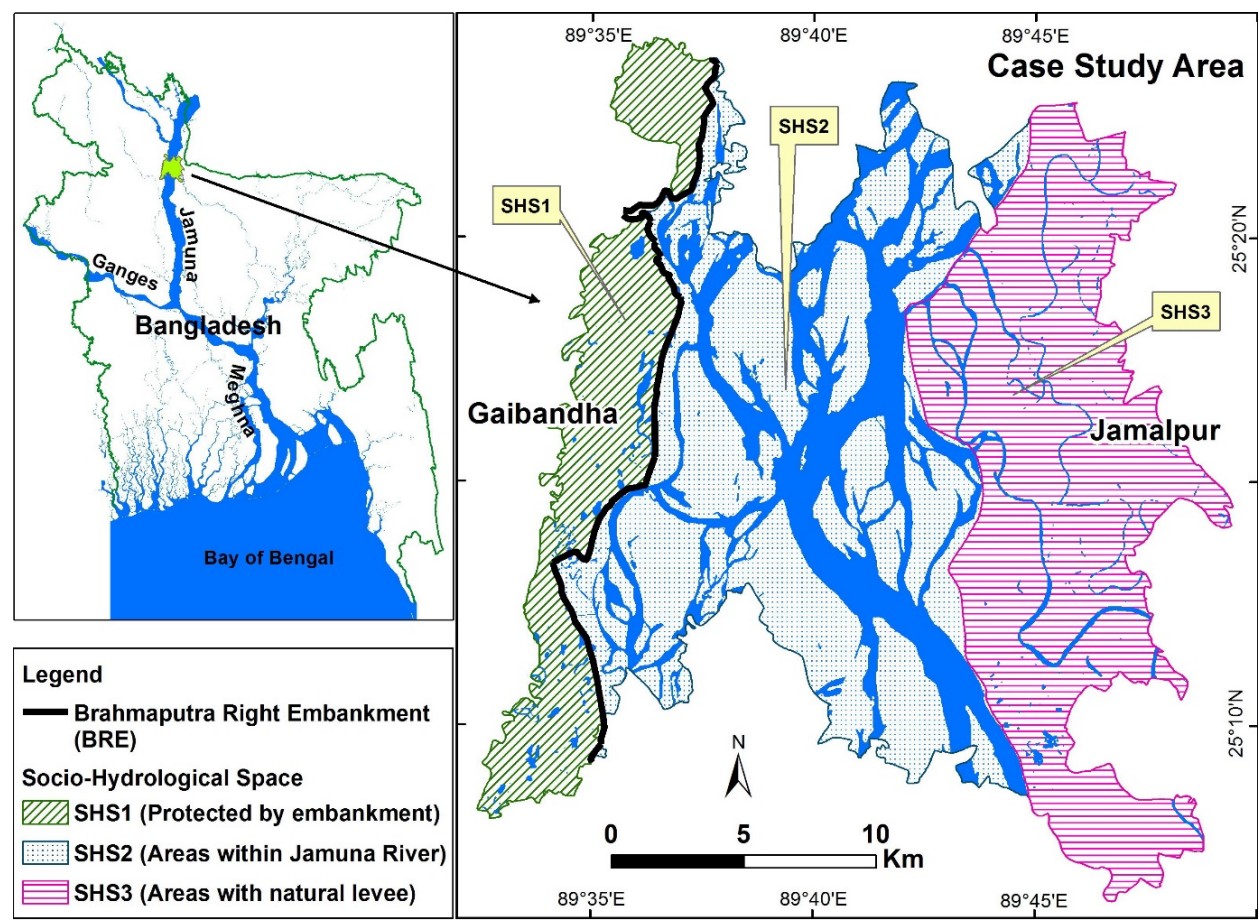

**Figure 2: Bangladesh map with case study area and SHS.**

## 4.2 Step 1: preliminary identification of SHS and classification of areas

Throughout the fieldwork period needed to collect the primary data described in Section 4.3 below, a detailed knowledge of
physical, technical and social conditions of the area was accumulated by the first author. In collecting this information, he
built upon and was guided by his personal knowledge as a resident in a nearby area, as well as by 10 years of professional
experience throughout Bangladesh as a water engineer charged with flood forecasting and training residents on using flood
and erosion forecasts. Since flood control measures were only developed along some rivers (see Section 4.1 above), the
study area is characterized by different degrees of protection. In addition to these human-made structures, different
geomorphological conditions influence local flood frequency and extent as well as the extent of river bank erosion.
Inhabitants adapt to these physical conditions, which is apparent e.g. in private investment levels and cropping patterns, but
also in public investment e.g. in schools and roads. These qualitative observations formed the basis for distinguishing three
SHS in the landscape, which are described in a narrative fashion in Section 5. To demarcate the SHS we used administrative

boundaries (unions and mauza) since this enabled the use of Government data in Step 2; 15 unions are included in the study area.

## 4.3 Step 2: evidence

The demarcation of SHS was validated through the analysis of primary data (household surveys and focus group
discussions) and secondary data (statistics, maps etc.) collected during the dry seasons of 2015 and 2016. The principal set of primary data consists of approx. 900 questionnaires dealing with several themes: general information (location of settlement and agricultural land, main occupation, age, income and expenditures, wealth and origin of the households), information on different flood experiences (depth of floods, frequency, duration, flood damages, effects on agricultural income and expenditures, adaptation options, migration etc.) and experiences with river erosion (frequency, damages, migration,
adaptation options etc.). We also set up focus group discussions in most unions in the case study area to validate and contextualize the survey data. Details of these methods are given below.

A cross-sectional method was used to gather the primary data of the case study area. Cross-sectional research involves using different groups of people, both male and female (farmer, fisherman, day-labour, service holder etc.) who differ in the variables of interest but share other characteristics, such as socio-economic status and ethnicity. We aimed to collect
approximately the same number of surveys in each of the three SHS. Due to the rural character, most of the respondents were farmers. We introduced an age bias because we wanted to collect historical information on flooding, riverbank erosion, livelihood etc. The household surveys were implemented with a combination of purposive sampling and quota sampling. Purposive sampling is a method where individuals are selected because they meet specific criteria (e.g., farmer, fisherman, day labour etc.). The quota sampling method selects a specific number of respondents with particular qualities (like farmer's
age should be 40 or above). We used the Raosoft sample size calculator to determine the required sample size for the surveys by union. This calculator allowed to enter values including acceptable margin of error, response distribution, confidence level and size of the population that is to be surveyed. We accepted a 5% margin of error with 95% confidence level to determine the sample size, which is 1% households (863 household surveys) of the case study area. The questionnaire for the survey is provided in the supplementary materials (ESM1).
In addition, we performed 12 focus group discussions in the case study area, four meetings in different unions in each SHS. About 20 participants were present in each of the meetings. Participants were selected based on occupation and location of the households, guaranteeing a uniform spread over the union area. The topics of the discussions were: how is flooding affecting livelihoods; what household coping strategies are used in relation to flooding, for example changing occupation or raising homesteads; migration patterns; community interventions against flooding; river bank erosion and household coping
strategies; community interventions against riverbank erosion; governmental initiatives against flooding and riverbank erosion etc. The agenda of the focus group discussions is provided in the supplementary materials (ESM2).

We also collected secondary data like time series satellite images to analyse the morphological dynamics of the Jamuna, census population data to analyse population density from different governmental and non-governmental organisations of Bangladesh. Results of Step 2 are discussed in Section 6.

## 5 Results step 1: Identification of socio-hydrological spaces along the Jamuna River

As noted, in our study area along the Jamuna three distinct socio-hydrological spaces were identified. SHS1 covers the areas protected by the BRE (on the west bank), SHS2 covers the char areas (in the river bed) and SHS3 includes areas with a natural levee (on the east bank). These are depicted in a schematized fashion in Fig. 3 and described by means of narratives below.

### 5.1 Areas protected with flood embankment (west bank) (SHS1)

This socio-hydrological space is protected from regular annual flooding, the so-called 'normal floods', by the embankment along the main river Jamuna (BRE) and along some smaller Jamuna tributaries. However, different parts of the area are still frequently inundated with excess rainwater, due to their low elevation and limited drainage capacity. Further, a few small rivers (Ghagot and Alai) inundate unprotected areas yearly in the western part of the area. Because the BRE effectively protects the area against all but the largest riverine flooding from the Jamuna, inhabitants feel confident enough to invest in
businesses and homesteads. In the study area, Gaibandha district, the BRE is not very well maintained, so the BRE sometimes breaches. Inhabitants build their houses on artificially raised platforms – often several metres above ground level – to reduce their vulnerability to the resulting floods. River bank erosion in this area is not widespread, but does occur in several locations. SHS1 therefore shows a combination of the 'fight' and 'adapt' patterns.

### 5.2 Floodplain outside the embankment (west bank) and chars (SHS2)

This is a very dynamic environment. The Jamuna is a braided river, where multiple channels crisscross within the outer boundary of the river. When considered over decades, the outer boundary is moving in a westward direction (CEGIS, 2007). The 'chars' – or river islands – are also moving, progressing or disappearing, due to local erosion processes. Chars have different ages, which have a direct relation to the height level. As the river still deposits sediment on chars, some older chars have higher elevations than the areas in SHS1, and have shown to remain dry in extreme flood conditions. If a newly
developed char does not erode immediately, it is first colonized by grass, which accelerates deposition of silt during the next flooding. Subsequently, people start to occupy the char, planting fast growing trees and laying out agricultural fields. In the course of time, all kind of facilities like schools, mosques, small shops, bazars etc. are established. Since the chars are not stable, most of the houses built in the chars are semi-permanent and easy to take apart and move. House types are kutcha (wood, straw and bamboo mats) or jhupri (straw). Many people raise the plinth levels of their houses to avoid flood
damages, but this is not very effective.

On the chars inhabitants regularly face damages from flooding and river bank erosion to agricultural land and crops and their homestead, often leading to complete destruction. Temporary migration during the flood season to safer places, for example the embankment or on railway lines, is therefore very common. Permanent resettlement occurs only when the land that people live and farm on simply disappears, although they usually find another place to settle on a nearby char when flood water have receded. People also sometimes change their occupation temporarily or permanently. As char dwellers' life styles are defined by flood and erosion, they appear to be able to cope with the harsh conditions. Yet, most of them become poorer through time, because of landlessness, unreliable and changing sources of employment, and frequent temporary migration or resettlement. SHS2 therefore shows only the 'adapt' pattern.

### 5.3 Eastbank (areas with natural levee) (SHS3)

The natural levee on the east bank of the Jamuna protects this area from about half of the annual riverine flooding; flooding occurs more frequently than in SHS1. A few areas are flooded by smaller rivers like the Old Brahmaputra and Jinjira. High water levels in these rivers sometimes occur independently of high water levels of the Jamuna, as these are not part of the same drainage basin. River bank erosion is conspicuous in this area. Even though the river as a whole shows a gradual westward shift, due to the presence of highly erodible bank materials on the left bank erosion is still severe in SHS3. For example, 75 ha of land eroded in 2015 in this area, of which 4 ha with housing (CEGIS, 2016). Inhabitants take the initiative to build small spurs and bank protection, made from bamboo and wood, to try to stop erosion. However, while these encourage sedimentation at a local scale, they are not sufficient to stop large scale erosion. As in SHS1, most houses are built on artificially raised mounds, substantially reducing potential for flood impacts. Flooding and riverbank erosion cause damage to agriculture, homesteads and businesses, in turn impoverishing people. As in SHS1 and SHS2, migration is one of the coping strategies, while households also adapt their cropping pattern to accommodate flooding and cultivate fast growing crops after the flood season. SHS3 therefore shows a combination of the 'fight' and 'adapt' patterns, with more 'adapt' and less 'fight' than SHS1.

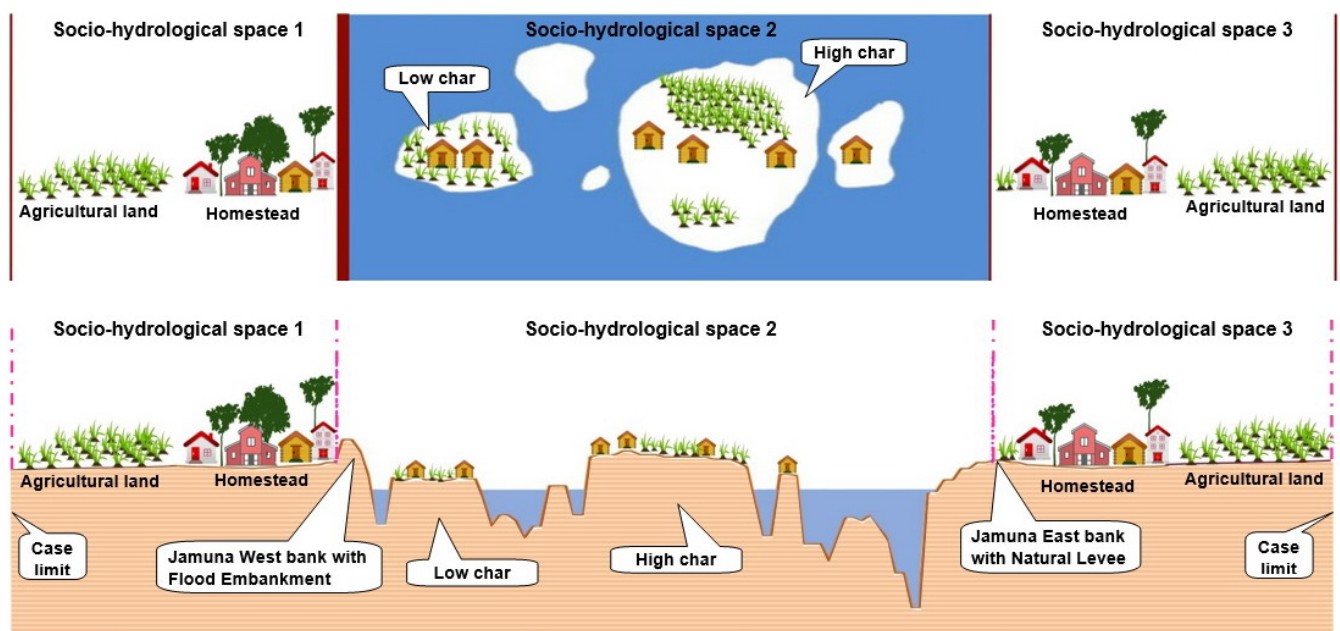

Figure 3: A typical planform and cross-section with distinct SHS along the Jamuna.

## 6 Results step 2: Evidence of socio-hydrological spaces along the Jamuna River

Using the data described in Section 4.3, in this section we show that the three SHS described above are significantly different. We only show the results for a limited number of variables: perceptions of the sources of flooding; flood frequency; flood damages; average household income and wealth; river bank erosion; migration; homestead types in the three identified SHS. We performed statistical analysis Chi-square test and ANOVA test ($p<0.05$) with these data for all analyses below (details are provided in ESM3). In each case the data for the three socio-hydrological spaces were significantly different.

### 6.1 Perception of the sources of flooding

All respondents have experienced flooding in their lifetime, but their perceptions about the sources of flooding are different (Fig. 4a). The main sources of flooding in space SHS1 are excessive rainfall, neighbouring small rivers and the Jamuna (through breaching of the BRE), whereas the sources of flooding mentioned in SHS2 is only Jamuna, and for SHS3 they are the Jamuna, the Old Brahmaputra River and other smaller rivers. A good number of people in SHS3 mentioned that the lack of embankment is one of the reasons for flooding, although they also mention excess discharges and river sedimentation.

## 6.2 Flood occurrence

When asked about their recollection of historical flood events (Fig. 4b), in SHS2 people indicated experiencing flooding every year. In both other spaces, this is roughly only once every 2 years. The unexpected relatively high flood frequency for the protected SHS1 may be attributed to the frequent failure of the embankment and to the fact that the area is flooded from the west by the Ghagot River, a tributary of the Jamuna.

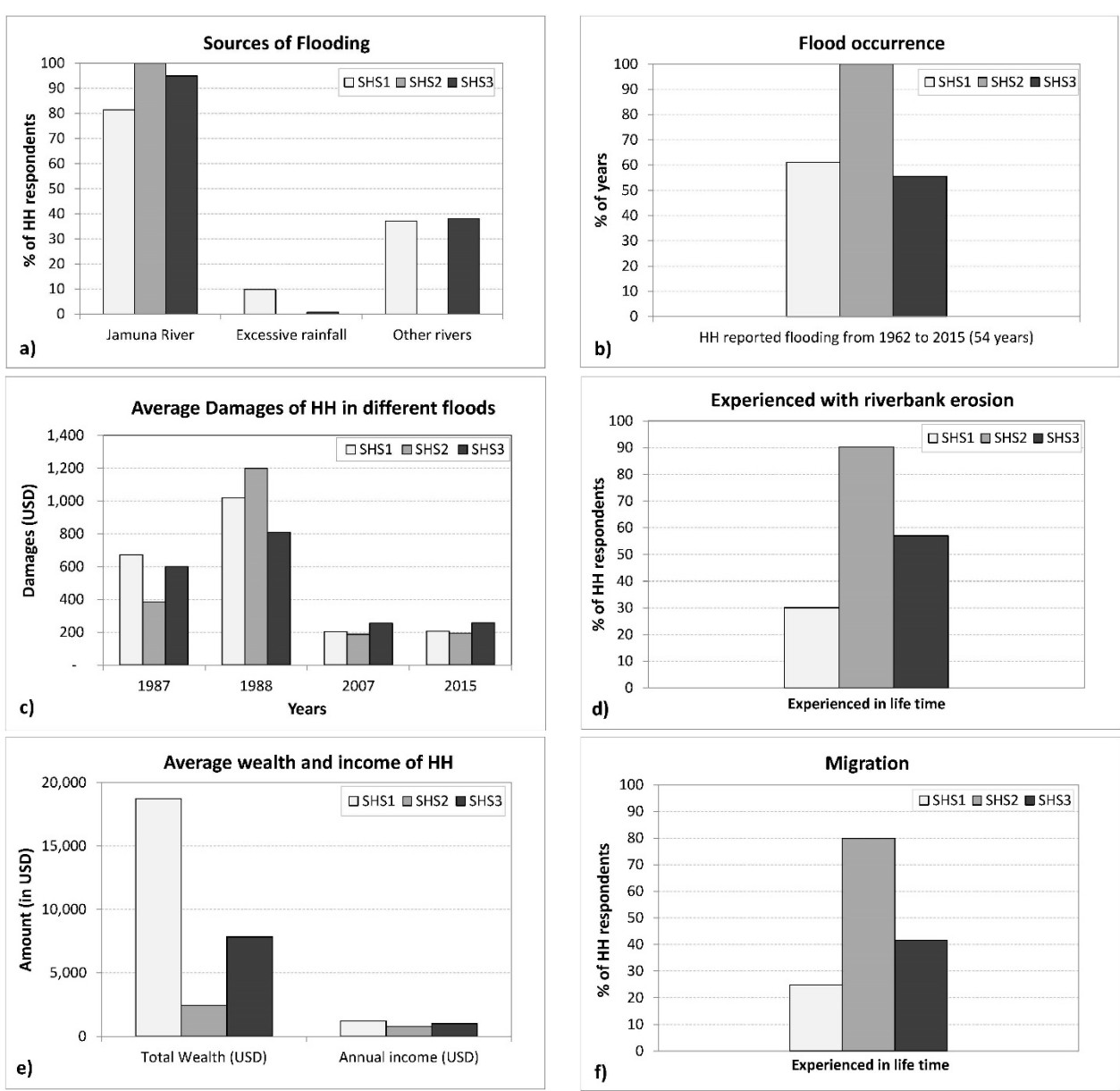

Figure 4: Comparison in between different socio-hydrological spaces (HH = household).

## 6.3 Flood damage

The 1988 flooding was the most severe event for all three spaces, ranging from an average damage of 800 USD per household in SHS3 to 1200 USD in SHS2 (Fig. 4c). In other years, average flood losses were much less. In 1987, damages in SHS1 were highest of the three spaces (~700 USD/household). This may be attributed to poor drainage capacity in SHS1, as well as a lower average land elevation, resulting in deeper and longer water logging. Damages in 2007 and 2015 show little difference between the three SHS (~200 USD/household). It is interesting to observe that (apart from the 1988 event), flood damage in SHS2 is lower than damage in SHS1 and SHS3. This is not only because people there are generally poorer (Fig. 4e), but also because people there are better adapted to frequent flooding (as they get flooded every year, see Fig. 4b). Yet, while people in SHS2 have adapted to frequent flood events, this adaptation does not make them less vulnerable to big floods, such as the one of 1988 (see Fig. 4c). This outcome was unexpected, and it would not be captured by any of the current models of human-flood interactions proposed so far.

## 6.4 River bank erosion

Riverbank erosion is experienced in each SHS, but (as expected) mainly by inhabitants in the dynamic SHS2 (Fig. 4d). However, erosion in SHS3 is also very high, with over 50% of the interviewed people having experienced it. In SHS1 expected rates are lowest, but still considerable, as 30% have experienced it due to breaching of the BRE.

## 6.5 Average household income and wealth

The average wealth distribution (Fig. 4e) shows clearly the economic differences between the households in the three SHS. In the protected areas, people have much more wealth, on average about 19,000 USD/household, against approx. 2,500 USD in SHS2 and 8,000 USD in SHS3. Household wealth includes land (homestead, agricultural, other land), ponds, houses and housing materials, livestock and portable wealth like savings, gold and silver. About 80% of the people in the case study area are farmers, so their income mostly depends on their agricultural production, complemented by remittances from migrant labour by family members for some families and from occasional day labour in agriculture, construction or fishing, or as rickshaw driver or van puller. Their starting position and subsequent losses depend to a large extent on where they live. The current situation is (much) worse for most households than in the past. As per our survey, in SHS1 there were 7% large farmer households (with land> 3 hectare) in 1960 but after consecutive flooding events, this was reduced to only 2% in 2015 (Fig. 5). Those who owned most land in the past (> 3 hectare) gradually saw a decline in their farm land to medium (1-2.99 hectare) or small (0.2-0.99 hectare), with some even becoming landless. There were only 16% landless households in 1960, but this increased to 28% in 2015.

In SHS2 and SHS3 a comparable pattern can be observed. The number of large farm households reduced from 18% to 1% and landless farmer households increased from 7% to 48% in SHS2. In SHS3 the proportion of large landowners reduced from 10% to 2% and that of landless farmers increased from 18% to 41%. More than 80% of the respondents from SHS2

reported that they could not recover from the losses due to flooding and riverbank erosion. Many of them have to change their occupation temporarily, and 3% of the respondents in SHS2 changed their occupation permanently from farmer to day labourer. This is less than the reduction in land ownership would suggest because landless farmers will try to rent land to be able to cultivate their own crops. If this is the case, they share crops with the land owners or pay a fixed amount per year.

There is a possibility that some respondents exaggerated reported losses in the hope that the research would help to mobilize funds. The focus group discussions clarified this issue. They revealed that cropping patterns in SHS1, SHS2 and SHS3 are different. Respondents in SHS1 are cultivating three crops per year. In SHS3 people used to cultivate three crops in the past, but due to flooding, they now cultivate either two crops or only one crop per year, only in the dry season after floods have subsided. From the survey data it appears that in SHS1 only 15% of the respondents changed cropping patterns between the
1960s and the 2010s, against 53% in SHS3 and 40% in SHS2. A very small number of people have changed land use completely, for example from agriculture to homestead, from low elevation land to high elevation land by filling silts, or from agriculture to fallow etc.

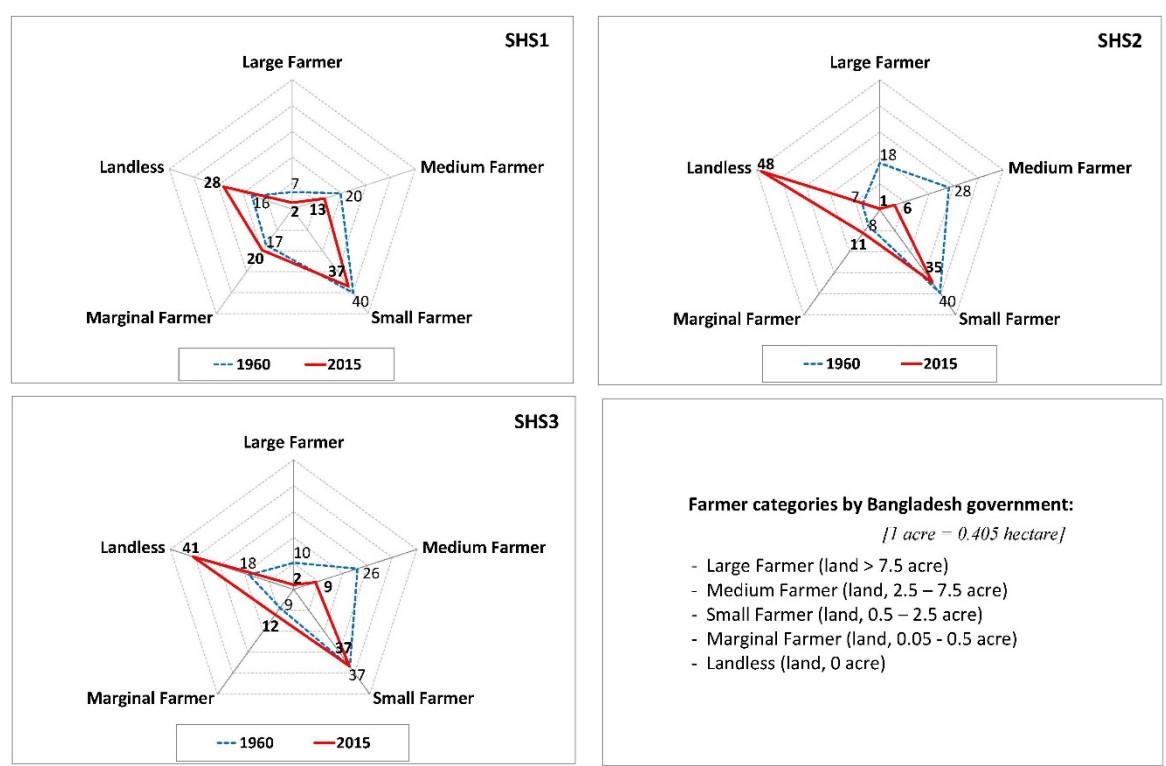

**Figure 5 Agricultural land changes with time of the different types of farmer (% of HH respondents).**

### 6.6 Migration

The population density in the three spaces from census data show much higher densities in SHS1 than in SHS2 and SHS3. In SHS1 it is 1,500 person per square km (varying between 1,000 to 3,000 person per square km in the different villages in SHS1), while population density in SHS3 is 800 person per square km (between 100 to 2,000 person per square km, the

lowest figure being for very few villages adjacent to the east bank). It is lowest in SHS2 at 400 person per square km (varying between 30 to 1,000 person per square km) (BBS, 2011). The historical population data from 1961 to 2011 show that population density has increased in most of the unions, except in SHS2 (CPP, 1961; BBS 1974, 1981, 1991, 2001 and 2011). Unfortunately, there are no official records of the exact number of people who migrate out of the area on a temporary

or permanent basis. From our survey, we found that temporary or permanent migration is most frequent in SHS2, mostly to SHS1 and SHS3. From 1988 to 2015, 17% of respondents had migrated to SHS1 and 8% to SHS3.

The study shows that riverbank erosion (Fig. 4d), more than flooding (Fig. 4b), is one of the main drivers for relocation from their place of origin (Fig. 4f). We found that 80% of the households in SHS2 had moved at least once. Most of them moved within 5 km, but in focus groups it was said that about 25% of people of that area had migrated away to other districts. About

68% of respondents were born in SHS1 and still live there, while 25% migrated to SHS1 from other places due to riverbank erosion. In SHS3, about 58% were born locally and the rest moved into the area, again mostly due to riverbank erosion. The respondents who relocated within the study area knew that their destination was flood prone and at risk from riverbank erosion. However, the lack of available land is a major problem so they contend with sub-optimal conditions.

### 6.7 Homestead types

The construction type of houses is different between the spaces (Fig. 6). Most of the pucca houses (well-constructed buildings using modern masonry materials) and semi-pucca or half pucca houses (made of brick and tin) are within SHS1 and SHS3, where people feel comparatively safe against flooding and erosion. As a result, they invest more in their home. In SHS2 a high proportion of kutcha (wood, straw and bamboo mats) and jhupri (straw) houses is observed, since these are easy to take apart and move in case of flooding or erosion, and less costly to construct.

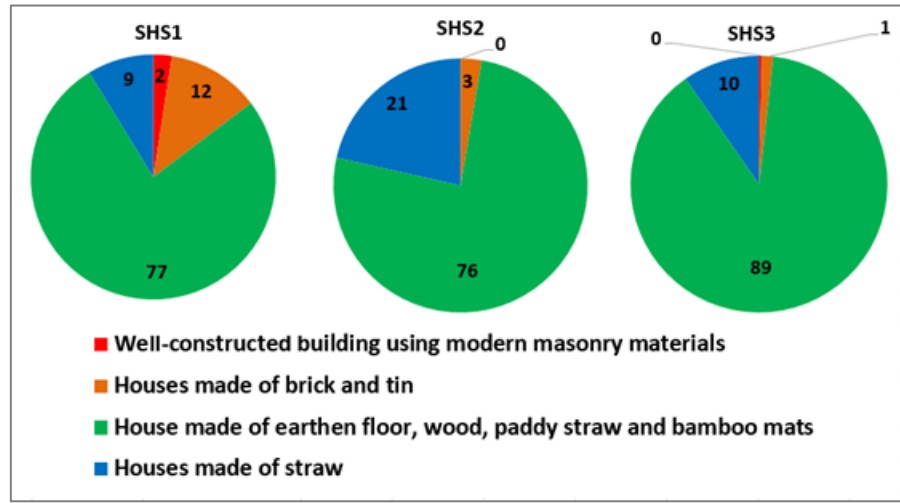

**Figure 6: Homestead type of households.**

# 7 Discussion

Based on thorough in-depth knowledge of the natural, technical and social conditions of the study area in the floodplain of the Jamuna River in Bangladesh, we proposed distinguishing between three SHS as the basic spatial units each with distinct socio-hydrological characteristics. Human-flood dynamics are different in each space, ranking from 'adapt to floods' (SHS3), to more (SHS1) or less (SHS2) 'fighting floods' in combination with 'adapt to floods' to the extent necessary. We then proceeded to demonstrate, through statistical analysis of primary and secondary data, that the SHS show significant differences in the following hydrological and social variables: perceptions of the sources of flooding, flood frequency, flood damages, average household income and wealth, river bank erosion, migration, and homestead types.

We thereby showed that there are good reasons to consider the three SHS as distinct both from a narrative and from a statistical perspective, and that such a distinction provides a good starting point for further socio-hydrological analysis of human-flood dynamics of the area. Further research can reveal more details of the socio-hydrological feedback loops and resulting patterns in data within the three SHS, for example by including urban areas. Our research is limited to rural inhabitants, and other patterns are likely to be revealed in SHS1 and SHS3 for urban areas, creating subdivisions within the spaces – or a reason to distinguish five instead of three spaces. The categorisation of the study area into spaces therefore depends on the focus of the study, but this does not invalidate the results. Rather, it shows that every abstraction, whether to find patterns or causal relationships, requires selective treatment of reality.

The issue of drawing boundaries around the SHS gives rise to another qualification. We started by outlining the boundaries of three SHS based on the presence of distinct physical features in the landscape: the embankment on the west bank, the natural levee on the east bank, and the riverbed in between. The exact boundaries were drawn on pragmatic grounds, using the administrative boundaries that best align with the physical features. These boundaries might show the approximate SHS in the present, but the boundaries of the physical and social systems are not fixed in time. The physical boundaries of the SHS are quite dynamic due to continuing bank erosion along both banks of the Jamuna (CEGIS, 2007). In particular, by analysing satellite images of the case study area from the late 1960s up to now, it appears that the west bank has been migrating westward and the east bank has been migrating eastward. As a result, the length-averaged width of the river has increased from 8.17 km to 11.68 km (CEGIS, 2007). Since the construction of the BRE in the 1960s, many breaches have occurred due to river bank erosion, forcing relocation of the embankment in many places (RBIP, 2015). At the same time due, to erosion of the east bank the natural levee also moved somewhat over time. Thus the physical boundaries between SHS1-SHS2 and SHS2-SHS3 are not fixed in time, while our statistical analyses assume that they are since they use the current physical boundaries. The social boundaries of the SHS in the Jamuna floodplain are also dynamic. Due to frequent relocations and migrations, the current inhabitants of the SHS may not have lived there throughout the study period since in every extreme event some migration occurs among the spaces (see Section 6.6). Therefore, the social boundaries between SHS1-SHS2 and SHS2-SHS3 are not fixed in time either, while our statistical analyses assume that they are since they relate to the current inhabitants. The dynamic nature of the boundaries of the SHS is unavoidable and indeed intrinsic to the highly

dynamic socio-hydrology of the floodplain system. It is therefore important to remember that the SHS are defined by their unique socio-hydrological characteristics compared with the surrounding area, not by their exact coordinates. For example, SHS2 is defined as a char within the river. If the river moves a kilometre and the char moves with it (or a different char forms), this does not change the definition of SHS2 as a char within the river. The same holds for the social boundaries, if one person moves to another SHS and adopts the strategies of that SHS, then the SHS does not change[1]. Ideally, the data collection and analyses of time series in Step 2 would follow these shifting boundaries, but this will most likely not be possible for due to data scarcity or time constraints.

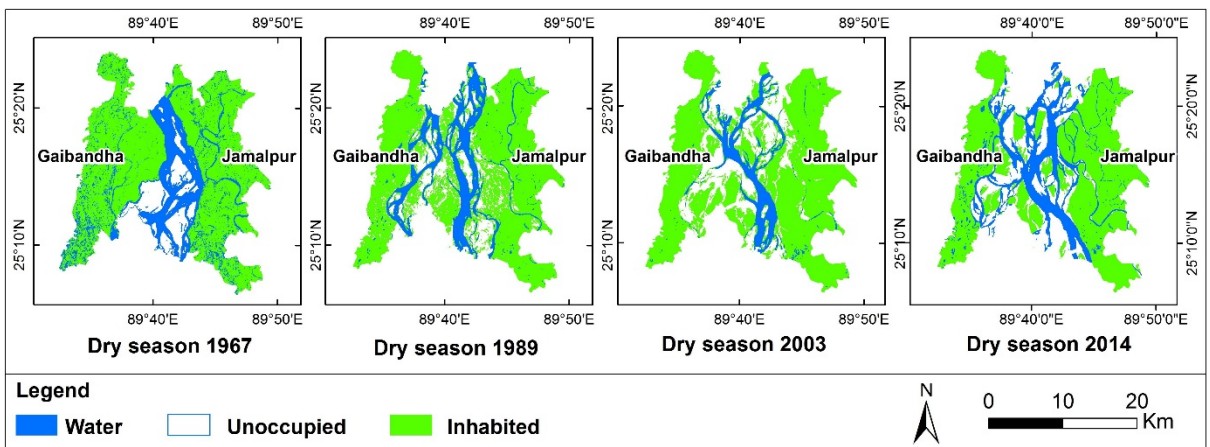

**Figure 7: Time series dry season satellite images of the case study area.**

## 8 Conclusions

We introduced the concept of socio-hydrological spaces (SHS) and applied it to a floodplain area along the Jamuna River in Bangladesh. SHS delineate areas where the interaction between social and hydrological processes show distinct characteristics, which in the case of floodplains can be classified as different combinations of two basic responses identified in the literature: 'fight floods' or 'adapt to floods'. SHS are therefore primarily a research tool that helps to identify patterns in a specific case. However, when SHS are applied to other floodplains this will enable global comparison of human-flood interactions elsewhere. For example, similar SHS to the ones found in the Jamuna floodplain are known to exist further down and upstream along the same river (known as Brahmaputra in India), so it would be worthwhile to compare socio-hydrological characteristics and analyse their differences and similarities.

Applying the SHS concept draws attention to the historical patterns in the co-evolution of social behaviour, natural processes and technological adoptions that give rise to different landscapes, different styles of living, and different ways of organizing livelihoods in specific geographical locations. The SHS concept suggests that the interactions between society and water are place-bound and specific because of differences in social processes, technological choices and opportunities, and

---

[1] We would like to thank one of the anonymous reviewers for helping us to formulate this insight.

hydrological dynamics. Such attention is useful anywhere in the world and also for other socio-hydrological systems than floodplains. It will be therefore be worthwhile to see whether SHS can also be used to analyse physical processes other than floods, such as droughts, salt intrusion, irrigated catchments or urban systems.

The usefulness of SHS does not only result from what it allows to see, as explained above, but also from the relative ease of application in situations where data are too sparse to use fully deterministic models (which is the case nearly anywhere in the world). Compared with existing approaches in socio-hydrology, the concept allows taking an intermediary (narrative and/or statistical) position between complex realities and generic models. As such, it is argued that SHS and generic models are complementary approaches with their respective advantages and disadvantages, making them useful for different purposes in different contexts.

Because SHS are place bound, and can only be found (literally) on the ground, the use of SHS forces the researcher to actually go to the field, talk to inhabitants and officials, and obtain a thorough understanding of the specifics of the location. This also means that the use of SHS will make socio-hydrological analyses more policy-relevant. In terms of practical use, it can be added as additional element to rapid rural appraisals, or other social assessments, to draw attention to how material conditions (hydrological and technical/infrastructure) co-shape social situations. The application of SHS is particularly useful to avoid broad-brush generalisations that do not take account of locality-bound problems due to the physical environment, without the need to interview every single household. SHS are therefore useful for developing interventions under disaster management, but also other development goals. In summary, SHS provides a new way of looking at and analysing socio-hydrological systems.

**Acknowledgements**

This research was funded by NWO-WOTRO grant W 07.69.110 'Hydro-Social Deltas: Understanding flows of water and people to improve policies and strategies for disaster risk reduction and sustainable development of delta areas in the Netherlands and Bangladesh'. Giuliano Di Baldassarre was supported by the European Research Council (ERC) within the project "HydroSocialExtremes: Uncovering the Mutual Shaping of Hydrological Extremes and Society" (ERC Consolidator Grant, No. 761678).

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
