# Peer review of "Socio-hydrological spaces in the Jamuna River floodplain in Bangladesh"

_Hydrology and Earth System Sciences, 2017_

## Referee Comment (RC1) · Anonymous Referee #1 · 25 Jan 2018

This is an interesting work based on a strong empirical and field based work. While I enjoyed reading the work, I was bothered by the concept of "socio-hydrological space" that the authors are pushing for. Why not just call it "social-hydrological system"? By calling it a "socio-hydrological space," what new things become possible that couldn't be achieved when you just simply call it socio-hydrological system? The notion of system has been around for long and it is exactly what the authors are trying to do. A system refers to an "integrated whole" and is composed of several interacting parts or elements. Of course, this assumes the presence of a boundary delineating which parts are inside the system and which are outside of it. And this boundary can be of different forms: spatial boundary, organizational boundary, ecological boundary, you name it. People specify these boundaries in an attempt to analyze and address specific research questions. So, system boundary is arbitrary and a system can be also nested within a higher level system. Let me challenge the authors. Can you define a larger socio-hydrological space that includes those three socio-hydrological spaces you described in the paper? I'm sure you could if you're comparing larger-level spaces between two very different regions. So, why not just use the term system? In social-ecology, they use the term "social-ecological system." They don't use "social-ecological space."

I also would like to see more discussion on how flood coping strategies vary by SHS1-SHS3. The authors do describe something, but not detailed enough. More details on how individual level strategies (cropping pattern, migration strategies, home flood-proofing) and group-level strategies (activities organized by communities) should be provided.

Figure 2 needs some improvement. Hard to see dotted line (levee). Hard to see boundaries of SHS1-3. If printed in B&W, these can't be distinguished.

I am also bothered by expressions like "adaptation space" and "levee effect space" in page 4. Adaptation and levee effect are emergent phenomena generated by system dynamics. I don't know what you mean by these can be rendered in terms of SHS.

Quite a few awkward grammars here and there. E.g., "channels more and more move into" (page 8).

In page 15, the authors say "the concept provides a methodological and theoretical advance in the socio-hydrology.." I am not convinced why this is so.

---

## Referee Comment (RC2) · Anonymous Referee #2 · 29 Jan 2018

Ferdous and colleagues developed a new concept called 'socio-hydrological spaces' which they define as a geographical area with distinct hydrological and social features that give rise to distinct patterns and emergent behavior. They then apply this concept to an analysis of the Jamuna River floodplain in Bangladesh. In case study they identify three distinct socio-hydrological spaces defined by geographical features and support this delineation with primary and secondary data. The example application is well supported by primary data collection. The application of mixed-method approaches is important in socio-hydrology and the topic is of interest to HESS readers. However, I do have a series of concerns that if addressed would strengthen the paper. I believe that with certain revisions it would be suitable for publication.

Comments

1. The definition of 'socio-hydrological spaces' hints at two different types of spaces. The first is space as a geographical area. The second is space as a portion of the parameter space which leads to a distinct set of emergent dynamics. (The examples of the adaptation space and levy effect space on page 4 further raise the question of the second type of space.) In the case presented, geographical features (e.g. embankment) are used to divide the case area into three sub-areas with different dynamics. Because these geographic features define the dynamics of the system all of the unions exhibiting similar dynamics are spatially clustered. However, I can envision cases in which the features defining the socio-hydrological dynamics are social not physical features. In these cases, I am not sure the 'spaces' would be contiguous. How would this approach be applied to a case where geographical features are poorly aligned with system dynamics? Or is this tool suitable for only the cases where geographical features are aligned with system dynamics?

2. In the definition section (pages 3-4), the authors present this concept/tool as an alternative to either narratives or mathematical models. However, in the case that follows the authors present both the 'socio-hydrological space' delineation with a case narrative, which I think was effective. Rather than serving as an effective standalone tool, 'socio-hydrological spaces' compliments these other approaches. I think the author's argument for this tool would be more convincing if they could frame it as part of a broad research plan. For example, the authors note that SHS is descriptive not explanatory. If combined with other approaches could it enhance the explanatory power of a study?

3. While it is important to expand the approaches used to address socio-hydrological questions and to synthesize quantitative and qualitative data, this is not the first study to do so. The authors should acknowledge other efforts in this space such as data-driven narratives (Treuer et al. 2017) and the pairing of statistical analysis and narratives (Hornberger et al. 2015), and articulate what 'socio-hydrological spaces' adds.

4. I think there is potential for this concept to be used comparatively across say multiple flood plain cases. Please speak to this potential.
5. Lastly, there are some typographic errors and awkward phrasing in the manuscript and it would benefit from a thorough review.

References

Hornberger GM, Hess DJ, Gilligan J (2015) Water conservation and hydrological transitions in cities in the United States. Water Resour Res 51(6):4635–4649.

Treuer G, et al. (2017) A narrative method for analyzing transitions in urban water management: The case of the Miami-Dade Water and Sewer Department. Water Resour Res 53(1):891–908.

---

## Referee Comment (RC3) · Anonymous Referee #3 · 30 Jan 2018

The authors aim to present a "new way of looking at and analysing socio-hydrological systems", and use a study area in the highly dynamic floodplains of the Jamuna river in Bangladesh.

After reading the introduction, I wanted to know: -how to construct and define a SHS -how the SHS improves or benefits the field of socio-hydrology, -how to apply the SHS to other research areas, or even to other areas within the country

General comments

I am impressed by the authors' knowledge of the study area. The methods used to construct the survey seem sound, the questionaires (ESM1) suitable to the research question at hand. The topic is of interest to HESS readers.

[Figure]

Overall, the paper seems to be a further development to the classification performed in Di Baldassarre 2013 and 2015 and rests on the assumption of two patterns of society-river interactions. (see also p. 5, line 1-5). While I understand that the concept of SHS is new to the field in terms of vocabulary, I don't see why the classification from Di Baldassarre 2015 which is criticized by the authors can not simply be performed on a smaller scale. Do we need SHS for that? How could the SHS concept be extended to the entire country? I am also not convinced by the results that this approach and the presented results draw "analytical attention to how flood dynamics co-evolve with societal dynamics".

My initial questions were only answered partly. I am unfortunately not sure how this method is an improvement or benefit to the field of socio-hydrology. I see the study's strong point in the extensive empirical field survey, but feel that this requires more work to show statistical relationships gathered from the individual SHS and then comparing those to hydrological data (flood extent, erosion, etc.). ). I also cannot easily detect how the SHS approach is useful in specifying the interaction between sociological and hydrological processes in the sense of the two-way feedbacks key to socio-hydrological approaches.

At present, SHS still seems to be a rather descriptive and classical approach to me, with the statistical methods mainly from the field of basic exploratory data analysis. While there is no harm in that, the authors do stress that they present a "new approach". The extensive surveys should be brought into context with actual observed data (especially Sec 5.1-5.4), in particular if the authors consider using this approach to make predictions (although it is unclear to me what they wish to predict and how, this is only mentioned in the beginning of the paper and should definitely be elaborated on). The authors should also adress the uncertainties in their work – there are a lot of biases inherent in conducting surveys, and I'm not sure the time span 1960-2016 is feasible due to the large number of external factors that could also contribute to e.g. migration or farm land area (such as the independence of Bangladesh in 1971).

[Figure]

Minor and major remarks

Section 3.1: Not entirely sure why this specific study area was chosen. How big are the invidiual SHS? Section 3.2: Why are socio-economic factors not relevant for the construction of the SHS? I would assume this makes a difference in how the livelihoods are affected by flooding in the individual SHS. (Add-on: on p.8, line 16 the occupation is liste das a delineating factor. I don't see this in section 3.2, where the delineation is based on "differences in geophysical characteristics and flood protection measures". Section 3.3: Is "evidence" the right word to use? Perhaps "data" is more suitable. Sections 4.1-4.3: can these be classified as Results? I would consider this to be part of the methodology/study area description. (VERY narrative) Section 4.2 Could benefit from references on chars. Sometimes, chars is in quotes, most of the time not. Please be consistent. Section 5.1 How was this verified? Using the household's answers can be deceptive, as there is a strong bias to the length of time since the last flood event. Also, of course a char in the Jamuna cannot be flooded by another river. Sections 5.4 Using only household surveys to state that e.g. "riverbank erosion is experienced in each zone", and to comment on how high these rates are without presenting physical observations is in my opinion not conclusive. I strongly suggest backing these statements up with observed erosion data. Also, how far away from the river do your respondents live? This can bias the answers, making the statements even more unconclusive. Section 5.7: how is this section relevant?

p. 3, line 24ff You state that the concept's importance lies "in its emphasis on how the interactions between society and wate rare always place-bound." Perhaps I use a different interpretation for the word place-bound, but the levee effect you mention afterwards is anything but place-bound. Rather, you describe yourself how this was introduced for the Po floodplain as well as by White in the US. Please clarify. P.5, line 3ff please clarify what you mean in this sentence – unclear to me. p.5, line 17: I am surprised that you do not mention any of the extreme flooding after 2007 – just last year severe flooding in the region occurred. On p.6, line 10 you do mention the

flood of 2016, so please check for consistency. Perhaps it would also be good to just name those years in which the study area was extremely flooded, not "general" extreme flood years in Bangladesh. p.6, line 11: how much percent was flooded? p.8, line 1: when were the focus group discussions with respect to the study years and the flood season? Also, during which season/months were the surveys conducted? p.8, line 10: Frequency analyses for what? The following sentence is unclear. p.8, line 26: how much of the bankline is eroded? p.9, line 29: migration to where? Outside of SHS3? p.10, line 17: is the unexpected flood frequency observed through e.g. data or satellite imagery? p.11, line 6: please include the flood damage information in the description of data and methods. How did you analyse what? How do the individual floods compare with respect to magnitude and flood duration in the individual SHS in each of those years? What about the study years? p.12, line 1: is there a citation for this? How low is the average elevation? It would be good to include this in the general description of the SHS. p.12, line 13: how ist he number of farmers with large households determined? If only from questionaire, how did you control for other biases such as migration, change of occupation? How certain do you think this number is? I would argue that the changes in SHS1 are not significant, and that they in particular cannot be attributed solely to consecutive flood events. Also, why 1960? Does it not make it more difficult to evaluate the results before/after Bangladesh became independent? When was the embankment in SHS1 built? Could the reduction of large farms not simply be due to other socio-economic developments in the region? Is this also solely based on information from questionaire? p.12, line 24: this is a major concern I also share in these types of studies (and I am not convinced this can be verified through a focus group discussion- how?). This is also why I stress the need for observed data. p. 13, line 7: increased by how much over what time period? p. 13, line 15: Did the respondents arrive or leave? The last two sentences of this paragraph are unclear to me. p. 14, line 11: which interactions between sociological and hydrological processes did you identify? Which two-way feedback are you refering to? p. 14, line 20: when do you consider the initial selection of the SHS to have "statistical meaning"? How is this

transferable? How can you be sure they are consistent over time? What is the added value of SHS if their boundaries are mobile? p. 15, line 18 ff: I agree. Please expand your methodology to include when your selected SHS need to be updated – for now, this is not quite clear. p. 15, line 22: what advance did you show? Which questions did you now answer that could not be answered before? How can you apply this in a broader sense? p. 16, line 16: To which policies, for example? What is a "rapid rural appraisal"?

Literature

Literature cited: the work largely cites and even uses figures from the same two papers (Di Baldassarre 2013a and b). While this is of course expected when developing the work of one research group further, what exactly is the point referencing literature such as the authors did in p.3, line 14 or p.4, line 15? I suggest to simply let the reader know where to look for the information or statement in the sentence before. FICHTER and nhc, 2015 does not look correct.

Language

Language is mostly good, but could definitely benefit from a careful read-through by a native speaker or a language editing service. E.g., p 2 Line 30 sees three uses of the word "different" in one sentence, and there are numerous grammatical or typographic mistakes throughout the paper. Be careful to introduce abbreviations before you use them (e.g. in abstract). p.5, line 27: "To evidence and understand…" sounds a bit awkward p.8, line 21: "inundated" instead of "ponded"? p. 15, line 14: what is "people mobility"?

Figures

Figure 2: cannot decipher the black names when printing out copy, perhaps resolution needs to be better. I had to look really closely to detect the boundaries oft he individual SHS. Why is the land colored red? Figure 4: please include the number of respondents

for each subset. Figure 5: Perhaps consider labeling all axes outside of plot or all inside plot (consistency). Also, it should say "SHS3". Starting when can a farmer be considered to sustain the own household? Figure 7: why is the land red? Why was the dry season chosen? A different coloring would greatly benefit the readability of the figure.

---

## Referee Comment (RC4) · Anonymous Referee #4 · 31 Jan 2018

The authors propose a new concept to study the interactions between humans and floods in a socio-hydrological system. They introduce the concept of Socio-Hydrological Spaces to describe a system that shows specific interactions between social, economic, hydrological, etc. factors that result in a certain behavior of the system and apply this to a case study in Bangladesh.

Although I can understand the advantages and potential of a comprehensive systematical approach to the study of "Socio-Hydrological Spaces" (which the authors seem to be aiming at) this new approach is quite poorly defined and explained. The authors merely give human-flood systems a different name (i.e. Socio-Hydrological spaces) and proceed to describe a case study as if this is a new approach. Mostert (2017) recently published an article in this same journal, arguing for case-study research as

an alternative approach for socio-hydrology and while his example of a case study is perhaps more qualitative than the one presented here, the authors should perhaps try to relate to his paper. Also, a very similar approach to the one presented in this manuscript for describing a case study of how humans and floods coexist, is presented by Hazarika et al. (2015).

The concept/approach would be new and in my opinion useful, if a general framework would be presented to analyze a case study/SHS in a comprehensive and consistent way, which would allow for the comparison of different Socio-Hydrological Spaces, their specific characteristics, and the feedbacks and phenomena that arise from the characteristics of this particular system. However, after reading the manuscript I did not really see how the method/concept that is presented here adds something new and useful to the already existing approach of a case study description.

Some more specific comments:

1) On page 2 in line 24-26 the authors state that "interactions and feedback mechanism between hydrological and social processes in floodplains remain largely unexplored and poorly understood" citing Di Baldassarre et al. 2013a. However, since this paper in 2013 there have actually been quite some studies that have explored these interactions (just a few examples: Viglione et al. 2014, Chen et al. 2016, Ciullo et al. 2017, etc.) and in fact the authors do acknowledge this later in the manuscript (page 3, line 5-6).

2) On page 2 the authors state that there are currently two approaches to socio-hydrology: qualitative studies and conceptual mathematical modelling studies. As I mention above, there are in fact other approaches (e.g. Mostert et al. 2017 and Hazarika et al. 2016) very similar to the approach that is presented here as a new approach.

3) The authors repeatedly state that running a conceptual mathematical model based on differential equations is much more data-demanding than the approach taken here. However, running a conceptual model like that does not require any data at all! Unless

one wants to compare the model with real data, which would indeed make it more data-demanding, but I would argue that it would be just as data-demanding as the approach taken here. In fact, in my opinion, using surveys and interview data is a very data-demanding approach (although a very valuable and useful approach).

4) In the discussion the authors state that the division into SHS and the testing is an iterative process. From the descriptions it seems that the "low char" and the "high char" are quite different from each other, so I wonder why the authors did not update their SHS based on the analysis?

5) In the discussion the authors state that: "Each SHS shows distinct features when comparing flood-society interactions, proving that the dynamic interactions of floods is dependent on different hydrological and societal characteristics along the Jamun River." The authors do indeed describe the different hydrological and societal characteristics of the three SHS, however, I miss the translation to the different dynamic interactions that follow from these characteristics. The description stops at describing the characteristics and does not describe the interactions and feedbacks that we are interested in in socio-hydrology. Are there in fact different ways of coping with floods in these three SHS? And if so, why do they behave differently? Which societal and hydrological combinations of characteristics lead to which kind of interactions? In the conclusion, the authors conclude that the concept draws attention to how historical patterns of co-evolution of social behavior, natural processes and technological adoptions give rise to different landscapes, different styles of living, and different ways of organizing livelihoods, while in fact the concept as it is presented here and applied to the case study, does not do this at all. It leaves me wondering what the different patterns, different styles of living, etc. are that emerged in these three SHS.

6) A large part of the discussion is about the spatial boundaries. The authors stress the point that the boundaries of the SHS move in time and that the physical boundaries between the three SHS are not fixed in time. While this is true, I do not really see why this is of importance. The SHS you define are defined by the characteristics of the

system, not by the exact coordinates. For example, the authors define SHS 2 as a char within the river, if the river moves a kilometer and the char moves with it (or a different char forms), this does not change the definition of SHS 2 as a char within the river. The same holds for the social boundaries, if one person moves to another SHS and adopts the strategies of that SHS, then the SHS does not change, does it? I think the authors could spend less attention on this in the discussion.

7) Figure 4 is not really consistent. The legend is placed in different locations, some graphs do show the total percentage on top of the bars and others don't (and some do but miss the %). Also, when printed in black and white, the difference between the color of SHS 1 and SHS 3 is not clear.

8) The format of figure 5 does not really allow for an easy comparison between the three SHS, I would suggest choosing another type of figure.

---

## Editor Comment (EC1) · M. Sivapalan (Editor) · 3 Feb 2018

The paper has received several critical but constructive comments. I strongly encourage the authors to respond to each reviewer, including an articulation of how they intend to address relevant comments and criticisms in a revised manuscript

---

## Short Comment (SC1) · 28 Feb 2018

A. Wesselink

a.wesselink@un-ihe.org

Reply to the comments of Anonymous Referee #1

We would like to thank our anonymous reviewer for his insightful and constructive comments. We apologize for our long silence; the lead authors were not aware of the HESS interactive method so we waited for all reviews to have been sent before replying. The comments from the reviewer have been reproduced in italic below, interspersed with our responses.

Referee comment: This is an interesting work based on a strong empirical and field based work. While I enjoyed reading the work, I was bothered by the concept of "socio-hydrological space" that the authors are pushing for. Why not just call it "socialhydrological system"? By calling it a "socio-hydrological space," what new things become possible that couldn't be achieved when you just simply call it socio-hydrological system? The notion of system has been around for long and it is exactly what the authors are trying to do. A system refers to an "integrated whole" and is composed of several interacting parts or elements. Of course, this assumes the presence of a boundary delineating which parts are inside the system and which are outside of it. And this boundary can be of different forms: spatial boundary, organizational boundary, ecological boundary, you name it. People specify these boundaries in an attempt to analyze and address specific research questions. So, system boundary is arbitrary and a system can be also nested within a higher level system. Let me challenge the authors. Can you define a larger socio-hydrological space that includes those three socio-hydrological spaces you described in the paper? I'm sure you could if you're comparing larger-level spaces between two very different regions. So, why not just use the term system? In social ecology, they use the term "social-ecological system." They don't use "social-ecological space."

Response: From all four reviewers' comments, we have come to the conclusion that the article in its current form does not yet convincingly define (and explain the need for) the concept of socio-hydrological spaces (SHS). We think SHS provides a methodological (and possibly paradigmatic) bridge between two contrasting approaches to studying human-water interactions: hydrosocial research based in sociology and human geography, and socio-hydrology based in hydrology and physical geography. These are described and discussed in Wesselink, A., Kooy, M. and Warner, J. (2017) "Socio-hydrology and hydrosocial analysis : toward dialogues across disciplines", WIREs Water 4(2) 1–14. Hydrosocial research take the messiness of the socionatural world as a given and results in location-specific narrative case study analyses with limited or no attempt at generalisation. Socio-hydrology looks to generalise findings from case studies through a system-approach using conceptual and mathematical models. "Socio-hydrological system" is thereby an abstract entity detached from the reality on the ground. We propose "socio-hydrological space" as a tool that helps to make the

necessary intermediary step between the messy reality of the specific location (space) and the abstract system of conceptual and mathematical models. The primary function of SHS is as a lens through which to view the complex reality of specific cases in order to find patterns in human-river interactions, which can then be compared to patterns in other locations to see if further generalisation towards universal models is possible. Its use invites the researcher to have an open mind to the existence of expected or unexpected patterns in location-specific data using a thorough understanding of the location: society, economics, natural system, technical interventions, etc. Subsequently, other cases may be analysed in order to explore whether the same or different patterns occur. These patterns can then be generalised through the more formal conceptualisation of socio-hydrological systems. On the one hand SHS thereby relates to a specific space, on the other hand it helps to find general patterns of human-river interactions: it serves as a methodological intermediary step or bridge between hydrosocial research and socio-hydrology. We are not familiar enough with SES research to be able to identify a similar concept that could be useful in SES research. Also, we are not aware that research on socio-ecological systems includes an alternative paradigm besides SES research but are happy to be informed differently. One reason for launching SHS is the existence of two research paradigms and our wish to bring these together; if SES research does not have a second paradigm then one of reasons for proposing SHS is thereby obsolete. The importance of such an intermediary step is illustrated by the differences between our findings on human-river relations in the Jamuna floodplain and those by Di Baldassarre et al. published in several papers for the Po valley. From Di Baldassarre et al.'s analysis of human-river relations in the Po valley it appears that two alternative responses exist in time and space (levees or adaptation). This same pattern would also be broadly recognisable in other high income countries where control of the river is a financial and technical possibility, such as The Netherlands (levees) or USA (some locations have levees, at others adaptation is required). However, society along the Jamuna show both responses at the same time in one region, but at different locations (SHS1 and SHS2), with a third intermediary response (SHS3). We

speculate that the greater variety in Bangladesh is due to less government budget and more difficult technical circumstances (the Jamuna is of a scale that renders most civil engineering works unsuccessful), but this remains for now an unexamined suggestion. If Di Baldassarre's findings are therefore taken to derive a general conceptual model for socio-hydrological systems along rivers, as in his subsequent publications with co-workers, the resulting models may be applicable to other rivers in similar conditions, but not to the Jamuna floodplain. Distinguishing socio-hydrological spaces in the field is therefore an important step in the search for generalisation of human-river interactions as they combine a place-based analysis with a presumption of the existence of generalisable patterns, without assuming that these patterns will be the same across the world. The proposition of using SHS to examine field data thereby also helps to overcome a bias towards high income, moderate climate regions in the study of (socio-) hydrology that was identified by James Linton (2008) in "Is the Hydrologic Cycle Sustainable? A Historical-Geographical Critique of a Modern Concept". Annals of the Association of American Geographers 98(3) 630-649. Regarding the question of boundaries, we agree that boundaries around a system are always arbitrary and selected in an attempt to analyse and address specific research questions, and a system can be also nested within a higher level system. However, the field data do suggest some boundaries as more logical or useful. In our case, the number of SHS that we found (three) is in first instance a result of the scale at which we explored the Jamuna human-river interactions (i.e. it is a result of the research scope/funding, not of the research question). However, we observe that the same pattern occurs along most of the Jamuna going downstream, until physical circumstances change too much and the river becomes tidal and under influence of cyclones. Going upstream, too, the pattern continues into India. While the three SHS we found are therefore first of all based on patterns in location-specific data, they can be generalised and used as a typology that can be applied elsewhere – but like the Po SHS they cannot be applied everywhere. It remains to be seen whether the same pattern of these three SHS occurs along other rivers and in other socio-economic conditions. Grouping the three SHS into one space

does not make sense because the three SHS describe three distinct human responses to distinct hydrological conditions; grouping them would eliminate the usefulness of the SHS concept for distinguishing between these different relations. Conversely, we will be publishing further research that finds differences in human-river interactions within SHS1, depending on the level of protection offered by the levee and the degree of urbanisation. These differences could be argued to constitute different SHS, but here it is the research objective that indeed determines whether further splitting up of one SHS is useful. So the scale of analysis to some extent determines the level of detail included in the SHS that are recognised, but not absolutely since merging the three SHS we distinguish does not make sense. However, patterns of SHS (such as the two options proposed by Di Baldassarre, or our three SHS) can be used to compare two different regions, as suggested by the reviewer. We could then find some regions where the options are similar to the Po valley, and other where they are similar to those in the Jamuna floodplain. And we think other patterns will exist. We contend that these patterns do not constitute (formal, mathematically conceptualised) systems, but this may be an matter of vocabulary only.

Referee comment: I also would like to see more discussion on how flood coping strategies vary by SHS1-SHS3. The authors do describe something, but not detailed enough. More details on how individual level strategies (cropping pattern, migration strategies, home floodproofing) and group-level strategies (activities organized by communities) should be provided.

Response: We recognise that more detail how flood coping strategies vary by SHS1-SHS3 is of interest. However, the purpose of this paper is to introduce the concept of SHS and provide illustrations of its use. Unfortunately there is no space in the current article to present all detailes research that we are conducting on human-rover interactions in the Jamuna floodplain. We are currently preparing a publication that addresses this topic in much more detail, including the historical developments that we cannot properly address here. We hope that reviewer is prepared to wait for this other

publication.

Referee comment: Figure 2 needs some improvement. Hard to see dotted line (levee). Hard to see boundaries of SHS1-3. If printed in B&W, these can't be distinguished.

Response: We will provide a clearer map in the revised manuscript.

Referee comment: I am also bothered by expressions like "adaptation space" and "levee effect space" in page 4. Adaptation and levee effect are emergent phenomena generated by system dynamics. I don't know what you mean by these can be rendered in terms of SHS.

Response: We agree that this terminology is not clear. We will reconsider this terminology in our revised paper.

Referee comment: Quite a few awkward grammars here and there. E.g., "channels more and more move into" (page 8).

Response: We apologize and will carefully review our language in the next version of the paper.

Referee comment: In page 15, the authors say "the concept provides a methodological and theoretical advance in the socio-hydrology." I am not convinced why this is so.

Response: We hope we have answered this concern in our reply to the first comment. In addition, because SHS are place bound, and can only be found (literally) on the ground, the use of SHS forces the researcher to actually go to the field, talk to inhabitants and officials, and obtain a thorough understanding of the specifics of the location. This also means that the use of SHS will make socio-hydrological analyses more policy-relevant. In terms of practical use, it can for instance be added as additional element to rapid rural appraisals, or other social assessments, to draw attention to how material conditions (hydrological and technical/infrastructure) co-shape social situations.

---

## Short Comment (SC2) · 28 Feb 2018

A. Wesselink

a.wesselink@un-ihe.org

Replies to Anonymous Referee #2

We would like to thank our anonymous reviewer for his insightful and constructive comments. We apologize for our long silence; the lead authors were not aware of the HESS interactive method so we waited for all reviews to have been sent before replying. The comments from the reviewer have been reproduced in italic below, interspersed with our responses.

Referee comment: Ferdous and colleagues developed a new concept called 'socio-hydrological spaces' which they define as a geographical area with distinct hydrological and social features that give rise to distinct patterns and emergent behavior. They

then apply this concept to an analysis of the Jamuna River floodplain in Bangladesh. In case study they identify three distinct socio-hydrological spaces defined by geographical features and support this delineation with primary and secondary data. The example application is well supported by primary data collection. The application of mixed-method approaches is important in socio-hydrology and the topic is of interest to HESS readers. However, I do have a series of concerns that if addressed would strengthen the paper. I believe that with certain revisions it would be suitable for publication. The definition of 'socio-hydrological spaces' hints at two different types of spaces. The first is space as a geographical area. The second is space as a portion of the parameter space which leads to a distinct set of emergent dynamics. (The examples of the adaptation space and levy effect space on page 4 further raise the question of the second type of space.) In the case presented, geographical features (e.g. embankment) are used to divide the case area into three sub-areas with different dynamics. Because these geographic features define the dynamics of the system all of the unions exhibiting similar dynamics are spatially clustered. However, I can envision cases in which the features defining the socio- hydrological dynamics are social not physical features. In these cases, I am not sure the 'spaces' would be contiguous. How would this approach be applied to a case where geographical features are poorly aligned with system dynamics? Or is this tool suitable for only the cases where geographical features are aligned with system dynamics?

Response: We indeed use the concept of SHS in the two ways suggested by the reviewer. We think SHS provides a methodological (and possibly paradigmatic) bridge between two contrasting approaches to studying human-water interactions: hydrosocial research based in sociology and human geography, and socio-hydrology based in hydrology and physical geography. These are described and discussed in Wesselink, A., Kooy, M. and Warner, J. (2017) "Socio-hydrology and hydrosocial analysis : toward dialogues across disciplines", WIREs Water 4(2) 1–14. Hydrosocial research take the messiness of the socionatural world as a given and results in location-specific narrative case study analyses with limited or no attempt at generalisation. Socio-hydrology

looks to generalise findings from case studies through a system-approach using conceptual and mathematical models. "Socio-hydrological system" is thereby an abstract entity detached from the reality on the ground. We propose "socio-hydrological space" as a tool that helps to make the necessary intermediary step between the messy reality of the specific location (space) studied by hydrosocial research and the abstract system of conceptual and mathematical models in socio-hydrology. The primary function of SHS is as a lens through which to view the complex reality of specific cases in order to find patterns in human-river interactions, which can then be compared to patterns in other locations to see if further generalisation towards universal models is possible. Its use invites the researcher to have an open mind to the existence of expected or unexpected patterns in location-specific data using a thorough understanding of the location: society, economics, natural system, technical interventions, etc. Subsequently, other cases may be analysed in order to explore whether the same or different patterns occur. These patterns can then be generalised through the more formal conceptualisation of socio-hydrological systems. On the one hand SHS thereby relates to a specific space, on the other hand it helps to find general patterns of human-river interactions by distinguishing different types of interactions, i.e. the second use of SHS as parameter space within all types of human-river interactions. The importance of such an intermediary step is illustrated by the differences between our findings on human-river relations in the Jamuna floodplain and those by Di Baldassarre et al. published in several papers for the Po valley. From Di Baldassarre et al.'s analysis of human-river relations in the Po valley it appears that two alternative responses exist in time and space (levees or adaptation). This same pattern would also be broadly recognisable in other high income countries where control of the river is a financial and technical possibility, such as The Netherlands (levees) or USA (some locations have levees, at others adaptation is required). However, society along the Jamuna show both responses at the same time in one region, but at different locations (SHS1 and SHS2), with a third intermediary response (SHS3). We speculate that the greater variety in Bangladesh is due to less government budget and more difficult technical circumstances (the Jamuna is

of a scale that renders most civil engineering works unsuccessful), but this remains for now an unexamined suggestion. If Di Baldassarre's findings are therefore taken to derive a general conceptual model for socio-hydrological systems along rivers, as in his subsequent publications with co-workers, the resulting models may be applicable to other rivers in similar conditions, but not to the Jamuna floodplain. Distinguishing socio-hydrological spaces in the field is therefore an important step in the search for generalisation of human-river interactions as they combine a place-based analysis with a presumption of the existence of generalisable patterns, without assuming that these patterns will be the same across the world. The proposition of using SHS to examine field data thereby also helps to overcome a bias towards high income, moderate climate regions in the study of (socio-) hydrology that was identified by James Linton (2008) in "Is the Hydrologic Cycle Sustainable? A Historical-Geographical Critique of a Modern Concept". Annals of the Association of American Geographers 98(3) 630-649. To use the concept of SHS empirically, we propose a two-step approach. First, a thorough understanding of a specific floodplain system (geography, history, technology, societal occupation etc.) results in a preliminary classification of the study area into distinct SHS. Second, the classification is tested for statistical significance using available or newly collected data. If the classification is not statistically significant, merging or splitting of categories should be considered where different social dynamics may be the reason for splitting (repeat step 1). The concept suggests that the interactions between society and water are place bound because of differences in social processes and river dynamics, but also generalisable since similar SHS patterns may be found elsewhere. Rather than a generalized model for understanding how such interactions occur, the concept draws analytical attention to how flood dynamics co-evolve with societal dynamics. Such attention is useful anywhere in the world and for other socio-hydrological systems than floodplains.

Referee comment: In the definition section (pages 3-4), the authors present this concept/tool as an alternative to either narratives or mathematical models. However, in the case that follows the authors present both the 'socio-hydrological space' delineation

with a case narrative, which I think was effective. Rather than serving as an effective standalone tool, 'socio-hydrological spaces' compliments these other approaches. I think the author's argument for this tool would be more convincing if they could frame it as part of a broad research plan. For example, the authors note that SHS is descriptive not explanatory. If combined with other approaches could it enhance the explanatory power of a study?

Response: We agree with the referee that SHS is complementary to narrative and mathematical approaches; in fact we believe it can (or even should) serve as a bridge between them, as we have explained in our previous answer. From a policy perspective, as we mentioned in our paper the distinction of SHS can for instance be added as additional element to rapid rural appraisals, or other social assessments, to draw attention to how material conditions (hydrological and technical/infrastructure) co-shape social situations. This would be useful for developing interventions under disaster management, but also other development goals.

Referee comment: While it is important to expand the approaches used to address socio-hydrological questions and to synthesize quantitative and qualitative data, this is not the first study to do so. The authors should acknowledge other efforts in this space such as data-driven narratives (Treuer et al. 2017) and the pairing of statistical analysis and narratives (Hornberger et al. 2015), and articulate what 'socio-hydrological spaces' adds.

Response: Thank you for pointing out some relevant narrative-cum-statistical studies that we should discuss. We will refer to these in our revised paper, with the caveat that these two papers discuss transitions in urban water management, which could be argued to relate to theory about socio-technical systems (as both papers acknowledge) with different drivers and conceptual models than those recognised in socio-hydrology research (see Van Staveren and Van Tatenhove, 2016: Hydraulic engineering in the social-ecological delta: understanding the interplay between social, ecological, and technological systems in the Dutch delta by means of 'delta trajectories'. Ecology and

[Figure]

Society 21(1):8. In fact, with SHS we are proposing a tool to help the comparison across cases which Treuer et al. (2015) identify as necessary next step: "Eventually, these narratives should be compared across cases". SHS offers the bridge between a specific case study, to identify patterns that can be compared to cases elsewhere.

Referee comment: I think there is potential for this concept to be used comparatively across say multiple flood plain cases. Please speak to this potential.

Response: We agree with you that there is potential for this concept to be used comparatively across multiple flood plain cases. We already referred to this in our answer to the first comment. In our case, the number of SHS that we found (three) is in first instance a result of the scale at which we explored the Jamuna human-river interactions (i.e. it is a result of the research scope/funding, not of the research question). However, we observe that the same pattern occurs along most of the Jamuna going downstream, until physical circumstances change too much and the river becomes tidal and under influence of cyclones. Going upstream, too, the pattern continues into India. While the three SHS we found are therefore first of all based on patterns in location-specific data, they can be generalised and used as a typology that can be applied elsewhere – but like the Po SHS they cannot be applied everywhere. It remains to be seen whether the same pattern of these three SHS occurs along other rivers and in other socio-economic conditions, and whether other SHS patterns exist in other floodplains.

Referee comment: Lastly, there are some typographic errors and awkward phrasing in the manuscript and it would benefit from a thorough review.

Response: We will make corrections of the errors in our revised manuscript.
* * *

---

## Short Comment (SC3) · 28 Feb 2018

**Replies to Anonymous Referee #3**

We would like to thank our anonymous reviewer for his insightful and constructive comments. We apologize for our long silence; the lead authors were not aware of the HESS interactive method so we waited for all reviews to have been sent before replying. The comments from the reviewer have been reproduced in italic below, interspersed with our responses.

Referee comment: The authors aim to present a "new way of looking at and analysing socio-hydrological systems", and use a study area in the highly dynamic floodplains of the Jamuna river in Bangladesh. After reading the introduction, I wanted to know: -how

to construct and define a SHS -how the SHS improves or benefits the field of sociohydrology, -how to apply the SHS to other research areas, or even to other areas within the country. I am impressed by the authors' knowledge of the study area. The methods used to construct the survey seem sound, the questionnaires (ESM1) suitable to the research question at hand. The topic is of interest to HESS readers. Overall, the paper seems to be a further development to the classification performed in Di Baldassarre 2013 and 2015 and rests on the assumption of two patterns of society river interactions. (see also p. 5, line 1-5). While I understand that the concept of SHS is new to the field in terms of vocabulary, I don't see why the classification from Di Baldassarre 2015 which is criticized by the authors cannot simply be performed on a smaller scale. Do we need SHS for that? How could the SHS concept be extended to the entire country? I am also not convinced by the results that this approach and the presented results draw "analytical attention to how flood dynamics co-evolve with societal dynamics".

Response: To use the concept of SHS empirically, we propose a two-step approach. First, a thorough understanding of a specific floodplain system (geography, history, technology, societal occupation etc.) results in a preliminary classification of the study area into distinct SHS. Second, the classification is tested for statistical significance using available or newly collected data. If the classification is not statistically significant, merging or splitting of categories should be considered where different social dynamics may be the reason for splitting (repeat step 1). From all four reviewers' comments, we have come to the conclusion that the article in its current form does not yet convincingly define (and explain the need for) the concept of socio-hydrological spaces (SHS). We think SHS provides a methodological (and possibly paradigmatic) bridge between two contrasting approaches to studying human-water interactions: hydrosocial research based in sociology and human geography, and socio-hydrology based in hydrology and physical geography. These are described and discussed in Wesselink, A., Kooy, M. and Warner, J. (2017) "Socio-hydrology and hydrosocial analysisr: toward dialogues across disciplines", WIREs Water 4(2) 1-14. Hydrosocial research take the messiness of the socionatural world as a given and results in location-specific narrative

case study analyses with limited or no attempt at generalisation. Socio-hydrology looks to generalise findings from case studies through a system-approach using conceptual and mathematical models. "Socio-hydrological system" is thereby an abstract entity detached from the reality on the ground. We propose "socio-hydrological space" as a tool that helps to make the necessary intermediary step between the messy reality of the specific location (space) and the abstract system of conceptual and mathematical models. The primary function of SHS is as a lens through which to view the complex reality of specific cases in order to find patterns in human-river interactions, which can then be compared to patterns in other locations to see if further generalisation towards universal models is possible. Its use invites the researcher to have an open mind to the existence of expected or unexpected patterns in location-specific data using a thorough understanding of the location: society, economics, natural system, technical interventions, etc. Subsequently, other cases may be analysed in order to explore whether the same or different patterns occur. These patterns can then be generalised through the more formal conceptualisation of socio-hydrological systems. On the one hand SHS thereby relates to a specific space, on the other hand it helps to find general patterns of human-river interactions: is serves as a methodological intermediary step or bridge between hydrosocial research and socio-hydrology. In one way, our classification is indeed a further development to the classification performed by Di Baldassarre 2013 and 2015, since it is based on two overall possibilities for response to flooding: protection or adaptation. However, in our case area we observed additional details in these responses that lead us to propose three human-river interactions rather than two. One bank is with man-made embankment and another bank is with natural levee. Because of the differential treatment of the right and left bank, and the existence of the chars in the middle of the river, three distinct system dynamic emerged. The third system is the char area which gained special attention in the SHS concept. The importance of using SHS as an intermediary step in the field data analysis is illustrated by the differences between our findings on human-river relations in the Jamuna floodplain and those by Di Baldassarre et al. published in several papers for the Po valley. From Di

СЗ

Baldassarre et al.'s analysis of human-river relations in the Po valley it appears that two alternative responses exist in time and space (levees or adaptation). This same pattern would also be broadly recognisable in other high income countries where control of the river is a financial and technical possibility, such as The Netherlands (levees) or USA (some locations have levees, at others adaptation is required). However, society along the Jamuna show both responses at the same time in one region, but at different locations (SHS1 and SHS2), with a third intermediary response (SHS3). We speculate that the greater variety in Bangladesh is due to less government budget and more difficult technical circumstances (the Jamuna is of a scale that renders most civil engineering works unsuccessful), but this remains for now an unexamined suggestion. If Di Baldassarre's findings are therefore taken to derive a general conceptual model for socio-hydrological systems along rivers, as in his subsequent publications with coworkers, the resulting models may be applicable to other rivers in similar conditions, but not to the Jamuna floodplain. Distinguishing socio-hydrological spaces in the field is therefore an important step in the search for generalisation of human-river interactions as they combine a place-based analysis with a presumption of the existence of generalisable patterns, without assuming that these patterns will be the same across the world. The proposition of using SHS to examine field data thereby also helps to overcome a bias towards high income, moderate climate regions in the study of (socio-) hydrology that was identified by James Linton (2008) in "Is the Hydrologic Cycle Sustainable? A Historical-Geographical Critique of a Modern Concept". Annals of the Association of American Geographers 98(3) 630-649. We argue that all spaces where humans interact with water are one or other type of SHS. However, not all SHS will occur in all places. The SHS concept suggests that the interactions between society and water are place bound because of differences in social processes and river dynamics, but also generalisable since similar SHS patterns may be found elsewhere. Rather than a generalized model for understanding how such interactions occur, the concept draws analytical attention to how flood dynamics co-evolve with societal dynamics. As for the Jamuna floodplain, Regarding the question of boundaries, we agree that boundaries around a system are always arbitrary and selected in an attempt to analyse and address specific research questions, and a system can be also nested within a higher level system. However, the field data do suggest some boundaries as more logical or useful. In our case, the number of SHS that we found (three) is in first instance a result of the scale at which we explored the Jamuna human-river interactions (i.e. it is a result of the research scope/funding, not of the research question). However, we observe that the same pattern occurs along most of the Jamuna going downstream, until physical circumstances change too much and the river becomes tidal and under influence of cyclones. Going upstream, too, the pattern continues into India. While the three SHS we found are therefore first of all based on patterns in location-specific data, they can be generalised and used as a typology that can be applied elsewhere - but like the Po SHS they cannot be applied everywhere. It remains to be seen whether the same pattern of these three SHS occurs along other rivers and in other socio-economic conditions. Patterns of SHS (such as the two options proposed by Di Baldassarre, or our three SHS) can be used to compare two different regions. We could then find some regions where the options are similar to the Po valley, and other where they are similar to those in the Jamuna floodplain. And we think other patterns will exist. We contend that these patterns do not constitute (formal, mathematically conceptualised) systems, but this may be an matter of vocabulary only. We argue that the concept "draws analytical attention" to how flood dynamics co-evolve with societal dynamics because the use of SHS forces the researcher to pay explicit attention to these interactions in his/her analysis.

Referee comment: My initial questions were only answered partly. I am unfortunately not sure how this method is an improvement or benefit to the field of socio-hydrology. I see the study's strong point in the extensive empirical field survey, but feel that this requires more work to show statistical relationships gathered from the individual SHS and then comparing those to hydrological data (flood extent, erosion, etc.). ). I also cannot easily detect how the SHS approach is useful in specifying the interaction between sociological and hydrological processes in the sense of the two-way feedbacks

**key to socio-hydrological approaches.**

Response: We hope we have now answered the reviewer's questions to a large extent in our reply to the first comment. Regarding statistical significance, it is not our goal to determine statistical relationships between physical and socio-economic data; we suspect that this will not yield significant correlations. However, we can show that individual variables are significantly different between the three spaces, which is what we do in our paper. Combining such statistics with narrative descriptions provides, in our view, enough evidence for the existence of three different SHS in the Jamuna floodplain. Finding statistical relationships for variables describing human-river interaction would be a next step towards the mathematical modelling practiced by socio-hydrology. We do not believe that such modelling is feasible in the reality of data scarcity and indeterminacy of relationships.

Referee comment: At present, SHS still seems to be a rather descriptive and classical approach to me, with the statistical methods mainly from the field of basic exploratory data analysis. While there is no harm in that, the authors do stress that they present a "new approach". The extensive surveys should be brought into context with actual observed data (especially Sec 5.1-5.4), in particular if the authors consider using this approach to make predictions (although it is unclear to me what they wish to predict and how, this is only mentioned in the beginning of the paper and should definitely be elaborated on). The authors should also address the uncertainties in their work – there are a lot of biases inherent in conducting surveys, and I'm not sure the time span 1960-2016 is feasible due to the large number of external factors that could also contribute to e.g. migration or farm land area (such as the independence of Bangladesh in 1971).

Response: The reason we label our proposal of SHS as 'new approach' is not based on the methods we use. They are indeed classical. The reason we present sociohydrological spaces as a new approach is because of the methodological innovation it entails. We think SHS provides a methodological bridge between two contrasting approaches to studying human-water interactions: hydrosocial research based in sociology and human geography, and socio-hydrology based in hydrology and physical geography. These are described and discussed in Wesselink, A., Kooy, M. and Warner, J. (2017) "Socio-hydrology and hydrosocial analysisr: toward dialogues across disciplines", WIREs Water 4(2) 1-14. Hydrosocial research take the messiness of the socionatural world as a given and results in location-specific narrative case study analyses with limited or no attempt at generalisation. Socio-hydrology looks to generalise findings from case studies through a system-approach using conceptual and mathematical models. "Socio-hydrological system" is thereby an abstract entity detached from the reality on the ground. We propose "socio-hydrological space" as a tool that helps to make the necessary intermediary step between the messy reality of the specific location (space) and the abstract system of socio-hydrological conceptual and mathematical models. The primary function of SHS is as a lens through which to view the complex reality of specific cases in order to find patterns in human-river interactions, which can then be compared to patterns in other locations to see if further generalisation towards universal models is possible. Its use invites the researcher to have an open mind to the existence of expected or unexpected patterns in location-specific data using a thorough understanding of the location: society, economics, natural system, technical interventions, etc. Subsequently, other cases may be analysed in order to explore whether the same or different patterns occur. These patterns can then be generalised through the more formal conceptualisation of socio-hydrological systems. On the one hand SHS thereby relates to a specific space, on the other hand it helps to find general patterns of human-river interactions: is serves as a methodological intermediary step or bridge between hydrosocial research and socio-hydrology. The importance of such an intermediary step is illustrated by the differences between our findings on human-river relations in the Jamuna floodplain and those by Di Baldassarre et al. published in several papers for the Po valley. From Di Baldassarre et al.'s analysis of human-river relations in the Po valley it appears that two alternative responses exist in time and space (levees or adaptation). This same pattern would also be broadly recognisable in other high income countries where control of the river is

a financial and technical possibility, such as The Netherlands (levees) or USA (some locations have levees, at others adaptation is required). However, society along the Jamuna show both responses at the same time in one region, but at different locations (SHS1 and SHS2), with a third intermediary response (SHS3). We speculate that the greater variety in Bangladesh is due to less government budget and more difficult technical circumstances (the Jamuna is of a scale that renders most civil engineering works unsuccessful), but this remains for now an unexamined suggestion. If Di Baldassarre's findings are therefore taken to derive a general conceptual model for sociohydrological systems along rivers, as in his subsequent publications with co-workers, the resulting models may be applicable to other rivers in similar conditions, but not to the Jamuna floodplain. Distinguishing socio-hydrological spaces in the field is therefore an important step in the search for generalisation of human-river interactions as they combine a place-based analysis with a presumption of the existence of generalisable patterns, without assuming that these patterns will be the same across the world. The proposition of using SHS to examine field data thereby also helps to overcome a bias towards high income, moderate climate regions in the study of (socio-) hydrology that was identified by James Linton (2008) in "Is the Hydrologic Cycle Sustainable? A Historical-Geographical Critique of a Modern Concept". Annals of the Association of American Geographers 98(3) 630-649. We disagree with the reviewer that surveys do not yield 'actual observed data', as implied in his/her advice that "extensive surveys should be brought into context with actual observed data". In sociological research surveys are an established way to collect objective data, especially when, as we did, protocols for representative sampling are followed. In fact, we contend that our survey data are at least as accurate as government statistics, which are also collected using the survey method. We refer to government statistics and other data such as satellite images where they are available, but there are indeed limitations on the availability and accuracy of such data for the whole period. For example, river level data are available at several sites, but cannot be related to inundation extent or depth since a detailed digital terrain model is not available. Also, population data are not available at the detailed level that we require. This is why we used the survey to collect interviewees' recollections of flooding, migration and flood damage. We addressed the uncertainty of getting accurate information from the surveys related to human recollection by arranging focus group discussions to minimize the errors. We do agree that uncertainties related to external factors like independence of Bangladesh in 1971 and its effect on migration is probably substantial, but this should affect the whole study region equally. More important uncertainty is introduced by shifting boundaries: as discussed in our paper, both the physical boundaries of the river bed and the human population move in time, which makes for unclear boundaries of the SHS. This issue is irresolvable and will need to be kept in mind with any interpretation of our results. Again, our aim is not to provide statistically significant correlations, but to present a lens with which to analyse field data to find patterns, which can be used in subsequent research to look for such correlations. Our future publications will include more details on historical developments, where we will make more extensive use of these data and combine then with government records where possible. We do not set out to be able to make predictions, and we apologise if the reviewer has interpreted our intentions in this way. Our aim is to better understand human-water interactions, and to provide a tool that helps to do this for concrete situations. By extension, we hope this this tool is also useful for policy-related studies since it highlights that there is no 'one size fits all' solution. We will make sure there is no doubt about our goals in the revised paper.

Referee's minor and major remarks: Section 3.1: Not entirely sure why this specific study area was chosen. How big are the individual SHS?

Response: We targeted the Januma river for this research because of the variety on flood management options and related human responses. We selected the upstream part north of the coastal zone because the coastal zone exhibits too many factors to be able to grapple with in this phase of research: there are tidal influences and cyclones and different stages of flood protection through polders. We have chosen this specific study area initially for the reason of being able to arrange access to the sites, i.e.

obtaining government permissions was relatively easy here because of personal ties to the area of the lead author. However, we assert that the same SHS also occur along much of the Jamuna upstream (in India) and downstream until the coastal zone in Bangladesh. The area of SHS1 is 74 km2, SH2 is 295 km2 and SHS3 is 126 km2; we will make sure to mention this is the revised paper. The number of SHS that we found (three) is in first instance a result of the scale at which we explored the Jamuna human-river interactions (i.e. it is a result of the research scope/funding, not of the research question). However, we observe that the same pattern occurs along most of the Jamuna going downstream, until physical circumstances change too much and the river becomes tidal and under influence of cyclones. Going upstream, too, the pattern continues into India. While the three SHS we found are therefore first of all based on patterns in location-specific data, they can be generalised and used as a typology that can be applied elsewhere – but like the Po SHS they cannot be applied everywhere. It remains to be seen whether the same pattern of these three SHS occurs along other rivers and in other socio-economic conditions.

Referee's minor and major remarks: Section 3.2: Why are socio-economic factors not relevant for the construction of the SHS? I would assume this makes a difference in how the livelihoods are affected by flooding in the individual SHS. (Add-on: on p.8, line 16 the occupation is listed as a delineating factor. I don't see this in section 3.2, where the delineation is based on "differences in geophysical characteristics and flood protection measures".

Response: During our delineation of SHS, we started with the physical and technical features in the landscape, and then found that socio-economic factors varied along with the physical ones, so they are definitely relevant. This sequence of defining a SHS is in accordance with the main direction of cause-effect relations in our case area: different SHS emerge due to differential degrees of protection. Theoretically, socio-economic factors could indeed be the logical starting point for the delineation of SHS in floodplain areas. We will make sure to remove the inconsistencies found by the reviewer in our

revised manuscript. We are currently exploiting our data further to investigate exactly this proposition. to be published at a later date. Within SHS1 the protection offered by the embankment varies according to rural/urban land use, and we are analysing whether this is a case of SHS being determined by socio-economic conditions.

Referee's minor and major remarks: Section 3.3: Is "evidence" the right word to use? Perhaps "data" is more suitable.

Response: We will replace "evidence" with "data".

Referee's minor and major remarks: Sections 4.1-4.3: can these be classified as Results? I would consider this to be part of the methodology/study area description. (VERY narrative)

Response: Section 4.1-4.3 are classified as Results because information within these sections are formulated from reconnaissance surveys and secondary data.

Referee's minor and major remarks: Section 4.2 Could benefit from references on chars. Sometimes, chars is in quotes, most of the time not. Please be consistent.

Response: We will be consistent in our reference to the chars in our revised manuscript.

Referee's minor and major remarks: Section 5.1 How was this verified? Using the household's answers can be deceptive, as there is a strong bias to the length of time since the last flood event. Also, of course a char in the Jamuna cannot be flooded by another river.

Response:. Sources of flooding resulted from household surveys and were verified with focus group discussions, since individual answers can indeed be deceptive. Further comments on our data collection methods were made in our response to the first comment.

Referee's minor and major remarks: Sections 5.4 Using only household surveys to

state that e.g. "riverbank erosion is experienced in each zone", and to comment on how high these rates are without presenting physical observations is in my opinion not conclusive. I strongly suggest backing these statements up with observed erosion data. Also, how far away from the river do your respondents live? This can bias the answers, making the statements even more inconclusive. Section 5.7: how is this section relevant?

Response: From time series satellite images it is possible to calculate the erosion in each spaces. CEGIS is doing this calculation for the whole Jamuna River every year. We mention the observed data in Sections 4.1-4.3 with the reference of CEGIS. People who are living near the banks or chars are experiencing with erosion and moving to other places. Though the river is very dynamic, many of them experienced erosion for several times. We captured their recollection using the surveys and focus group discussions. Regarding Section 5.7: depending on socio-economic conditions and flood damages, people's investments in their homestead is different. Homestead patterns are therefore an economic marker for human-river interactions in different spaces.

Referee's minor and major remarks: p. 3, line 24ff You state that the concept's importance lies "in its emphasis on how the interactions between society and water are always place-bound" Perhaps I use a different interpretation for the word place-bound, but the levee effect you mention afterwards is anything but place-bound. Rather, you describe yourself how this was introduced for the Po floodplain as well as by White in the US. Please clarify.

Response: The levee effect is a phenomenon that can indeed be observed to some extend across the world where society has decided to protect itself against flooding. It is the proposition of this paper that this phenomenon, in interaction with society, in turn gives rise to different socio-hydrological spaces in different places. SHS are thereby place-bound, but they also show patterns which can be turned into a typology, as we explained above. We will elaborate this dual character of SHS better in the revised paper.

Referee's minor and major remarks: P.5, line 3ff please clarify what you mean in this sentence – unclear to me.

Response: We agree with you that the sentence is not clear. We will rewrite the sentence as "We illustrate how the concept can be used in a more detailed way by doing refined analysis for the Jamuna floodplain in Bangladesh".

Referee's minor and major remarks: p.5, line 17: I am surprised that you do not mention any of the extreme flooding after 2007 - just last year severe flooding in the region occurred.

Response: We will add references in our revised manuscript.

Referee's minor and major remarks: On p.6, line 10 you do mention the flood of 2016, so please check for consistency. Perhaps it would also be good to just name those years in which the study area was extremely flooded, not "general" extreme flood years in Bangladesh. p.6, line 11: how much percent was flooded?

Response: We will ensure consistency in the revised manuscript and update the record. From household surveys we observed that our case area's extreme flood years were similar with the general extreme flood years in Bangladesh. Due to embankment breaching 100% of SHS1 was flooded in 2016.

Referee's minor and major remarks: p.8, line 1: when were the focus group discussions with respect to the study years and the flood season? Also, during which season/months were the surveys conducted? p.8, line 10: Frequency analyses for what? The following sentence is unclear.

Response: The focus group discussions were conducted in the January to April of 2016 and November of 2016. The household survey were conducted from October 2015 to April 2016 and September 2016 to December 2016. We will add this to the revised manuscript. We did flood frequency analysis. We will add the word "flood" in our revised manuscript. We also performed statistical analysis to check for relevance

of the differences between the three SHS on all data we used for the analysis.

Referee's minor and major remarks: p.8, line 26: how much of the bankline is eroded?

Response: The rate of bankline erosion is not fixed in the area. Along the right bank erosion started to reduce from the early 1990s. Erosion rate varies from 800 to 2,000 ha per year (CEGIS, 2007). We will add this information to our revised manuscript.

Referee's minor and major remarks: p.9, line 29: migration to where? Outside of SHS3?

Response: From surveys it is observed that most the people are migrating within the spaces. Very few of them are migrating outside SHS3 because they do not have access to land there. We will specify this in out revised manuscript.

Referee's minor and major remarks: p.10, line 17: is the unexpected flood frequency observed through e.g. data or satellite imagery?

Response: This information was obtained from our survey data. We have collected flood information from about 900 household respondents. All the respondent from SHS2 confirmed that they have faced flooding in every year while the respondents from SHS1 and SHS3 have informed us that they have faced flooding once in two years. The flood frequency (Figure 4b) is obtained from our survey data. We have also asked these information in the focus group discussion meetings to verify the frequency.

Referee's minor and major remarks: p.11, line 6: please include the flood damage information in the description of data and methods. How did you analyse what? How do the individual floods compare with respect to magnitude and flood duration in the individual SHS in each of those years? What about the study years?

Response: We will include the flood damage information in the description of data and methods in our revised manuscript. We have collected their assessment of damages for all the flood years that the respondents could remember. We analyse the average damages of households per year. Flood durations are not same for all years or in whole

area. The questionnaire was made available as supplementary information to provide all details on the data that was collected.

Referee's minor and major remarks: p.12, line 1: is there a citation for this? How low is the average elevation? It would be good to include this in the general description of the SHS.

Response: The value we showed in the text is produced from our data analysis. We will include this in the general description of the SHS in our revised manuscript.

Referee's minor and major remarks: p.12, line 13: how is the number of farmers with large households determined? If only from the questionnaire, how did you control for other biases such as migration, change of occupation? How certain do you think this number is? I would argue that the changes in SHS1 are not significant, and that they in particular cannot be attributed solely to consecutive flood events. Also, why 1960? Does it not make it more difficult to evaluate the results before/after Bangladesh became independent? When was the embankment in SHS1 built? Could the reduction of large farms not simply be due to other socio-economic developments in the region? Is this also solely based on information from questionnaire?

Response: Farm sizes are determined according to the classification by the government of Bangladesh. Details of farm sizes are available in "Census of Agriculture 2008, National series, Volume-1, Bangladesh Bureau of Statistics, published in November 2010, but these do not present the level of detail that we need. We collected data about migration and change of occupation at household level and also try to minimize error trough focus group discussions. The changes in agricultural farm land in SHS1 is indeed not significant while it is significant in SH2 and SH3. We took 1960 as starting point because the embankment was constructed in 1960s. Change in agricultural farms are solely based on information from household surveys. We will make sure to mention these details in our revised paper. We do agree that uncertainties related to external factors like independence of Bangladesh in 1971 and its effect on migration is

probably substantial, but this would affect the whole study region, though probably not to the same degree. The same applies to the reduction in farm size: due to growing population, the available land per family has reduced everywhere. We will make sure to mention these caveats in the revised paper.

Referee's minor and major remarks: p.12, line 24: this is a major concern I also share in these types of studies (and I am not convinced this can be verified through a focus group discussion- how?). This is also why I stress the need for observed data.

Response: As indicated above, we disagree with the reviewer that surveys do not yield 'actual observed data'. In sociological research surveys are an established way to collect objective data, especially when, as we did, protocols for representative sampling are followed. In fact, we contend that our survey data are at least as accurate as government statistics, which are also collected using the survey method, and definitely more detailed and extensive than the statistics that are available from the government The large number of surveys and the objective sampling method do compensate to a considerable extent for uncertainties in individual responses.

Referee's minor and major remarks: p. 13, line 7: increased by how much over what time period?

Response: Population density has increased from 600 to 1500 person per km2 in SHS1 during the period of 1960 to 2011, while in SHS2 it is from 200 to 400 person per km2 and SHS3 it is from 330 to 800 person per km2. We will make sure to include these details in our revised manuscript.

Referee's minor and major remarks: p. 13, line 15: Did the respondents arrive or leave? Response: In this case the respondents have arrived. We will make sure to clarify this in the revised paper.

Referee's minor and major remarks: The last two sentences of this paragraph are unclear to me. p. 14, line 11: which interactions between sociological and hydrological

processes did you identify? Which two-way feedback are you referring to?

Response: The hydrological processes we have identified are flooding and river bank erosion and the social processes are migration, livelihood changes of the households. We will mention this as such in the revised manuscript.

Referee's minor and major remarks: p. 14, line 20: when do you consider the initial selection of the SHS to have "statistical meaning"? How is this transferable? How can you be sure they are consistent over time? What is the added value of SHS if their boundaries are mobile?

Response: Because the identified SHS show statistically significant differences when selected variables are compared we call the SHS to have "statistical meaning". The boundary of the SHS are not consistent over time as the Jamuna River is very dynamic and erosion occurs in almost every year. This is a limitation for delineating the SHS boundaries. However, the main purpose of proposing and using the concept of SHS is not to define once and for all the exact boundaries of the SHS. Its main purpose lies in providing a lens through which to analyse the complex reality of human-river interactions, and to find patterns in this that can help to understand these relationships. Regarding the question of boundaries, in research boundaries around a system are always arbitrary and selected in an attempt to analyse and address specific research questions, and a system can be also nested within a higher level system. However, the field data do suggest some boundaries as more logical or useful. In our case, the number of SHS that we found (three) is in first instance a result of the scale at which we explored the Jamuna human-river interactions (i.e. it is a result of the research scope/funding, not of the research question). Patterns of SHS (such as the two options proposed by Di Baldassarre, or our three SHS) can be used to compare two different regions, as suggested by the reviewer. We could then find some regions where the options are similar to the Po valley, and other where they are similar to those in the Jamuna floodplain. And we think other patterns will exist. We contend that these patterns do not constitute (formal, mathematically conceptualised) systems, but this

may be an matter of vocabulary only. Put differently, in the words proposed by reviewer #4: the SHS we define are defined by the characteristics of the system, not by the exact coordinates. For example, SHS 2 is define as a char within the river; if the river moves a kilometre and the char moves with it (or a different char forms), this does not change the definition of SHS 2 as a char within the river, it just changes its location. The same holds for the social boundaries: if one person moves to another SHS and adopts the strategies of that SHS, then the SHS does not change.

Referee's minor and major remarks: p. 15, line 18 ff: I agree. Please expand your methodology to include when your selected SHS need to be updated – for now, this is not quite clear.

Response: We could envisage that the description/typology of the three SHS need to be changed when one of the main subsystems changes extensively. For example, if major flood management measures are brought in, if the river course changes suddenly, if economic development enables most char dwellers to find work elsewhere. Otherwise, the exact boundaries are not a major concern – see our previous reply.

Referee's minor and major remarks: p. 15, line 22: what advance did you show? Which questions did you now answer that could not be answered before? How can you apply this in a broader sense?

Response: We hope that we have replied to this comment by our extensive replies above. We will summarise these arguments in the conclusions. As to the broader use of SHS, The use of SHS forces the researcher to actually go to the field, talk to inhabitants and officials, and obtain a thorough understanding of the specifics of the location. This also means that the use of SHS will make socio-hydrological analyses more policy-relevant by highlighting that there are no 'one-size fits all' solutions. In terms of practical use, it can for instance be added as additional element to rapid rural appraisals, or other social assessments, to draw attention to how material conditions (hydrological and technical/infrastructure) co-shape social situations.

Referee's minor and major remarks: p. 16, line 16: To which policies, for example? What is a "rapid rural appraisal"?

Response: The policy could relate to disaster management, migration, flood management. Unfortunately, there is no generally accepted definition of rapid rural appraisal. Rapid rural appraisal is most commonly described as a systematic, semi-structured activity out in the field by a multidisciplinary team and is designed to obtain new information and to formulate new hypotheses about rural life in an intensive, short campaign.

Referee's minor and major remarks: Literature cited: the work largely cites and even uses figures from the same two papers (Di Baldassarre 2013a and b). While this is of course expected when developing the work of one research group further, what exactly is the point referencing literature such as the authors did in p.3, line 14 or p.4, line 15? I suggest to simply let the reader know where to look for the information or statement in the sentence before. FICHTER and nhc, 2015 does not look correct.

Response: we will simplify the references as requested by the reviewer. We will correct the error in the reference to FICHTNER and nhc (2015) Morphology: Feasibility Report and Detailed Design Priority Reach: Final Report, Annex A, Vol 1 River Bank Improvement Program.

Referee's minor and major remarks: Language is mostly good, but could definitely benefit from a careful read-through by a native speaker or a language editing service. E.g., p 2 Line 30 sees three uses of the word "different" in one sentence, and there are numerous grammatical or typographic mistakes throughout the paper. Be careful to introduce abbreviations before you use them (e.g. in abstract). p.5, line 27: "To evidence and understand: " sounds a bit awkward p.8, line 21: "inundated" instead of "ponded"? p. 15, line 14: what is "people mobility"?

Response: We will do a detailed revision of the text to eliminate grammatical and typographic mistakes in our revised manuscript.

Referee's minor and major remarks: Figure 2: cannot decipher the black names when printing out copy, perhaps resolution needs to be better. I had to look really closely to detect the boundaries of the individual SHS. Why is the land colored red? Figure 4: please include the number of respondents for each subset. Figure 5: Perhaps consider labeling all axes outside of plot or all inside plot (consistency). Also, it should say "SHS3". Starting when can a farmer be considered to sustain the own household? Figure 7: why is the land red? Why was the dry season chosen? A different coloring would greatly benefit the readability of the figure.

Response: We will replace the figures in our revised manuscript.

---

## Short Comment (SC4) · 28 Feb 2018

A. Wesselink

a.wesselink@un-ihe.org

Replies to Anonymous Referee #4 We would like to thank our anonymous reviewer for his insightful and constructive comments. We apologize for our long silence; the lead authors were not aware of the HESS interactive method so we waited for all reviews to have been sent before replying. The comments from the reviewer have been reproduced in italic below, interspersed with our responses.

Referee comment: The authors propose a new concept to study the interactions between humans and floods in a socio-hydrological system. They introduce the concept of Socio- Hydrological Spaces to describe a system that shows specific interactions between social, economic, hydrological, etc. factors that result in a certain behaviour

[Figure]

of the system and apply this to a case study in Bangladesh. Although I can understand the advantages and potential of a comprehensive systematic approach to the study of "Socio-Hydrological Spaces" (which the authors seem to be aiming at) this new approach is quite poorly defined and explained. The authors merely give human-flood systems a different name (i.e. Socio-Hydrological spaces) and proceed to describe a case study as if this is a new approach. Mostert (2017) recently published an article in this same journal, arguing for case-study research as an alternative approach for socio-hydrology and while his example of a case study is perhaps more qualitative than the one presented here, the authors should perhaps try to relate to his paper. Also, a very similar approach to the one presented in this manuscript for describing a case study of how humans and floods coexist, is presented by Hazarika et al. (2015). The concept/approach would be new and in my opinion useful, if a general framework would be presented to analyze a case study/SHS in a comprehensive and consistent way, which would allow for the comparison of different Socio-Hydrological Spaces, their specific characteristics, and the feedbacks and phenomena that arise from the characteristics of this particular system. However, after reading the manuscript I did not really see how the method/concept that is presented here adds something new and useful to the already existing approach of a case study description.

Response: From all four reviewers' comments, we have come to the conclusion that the article in its current form does not yet convincingly define (and explain the need for) the concept of socio-hydrological spaces (SHS). We think SHS provides a methodological (and possibly paradigmatic) bridge between two contrasting approaches to studying human-water interactions: hydrosocial research based in sociology and human geography, and socio-hydrology based in hydrology and physical geography. These are described and discussed in Wesselink, A., Kooy, M. and Warner, J. (2017) "Socio-hydrology and hydrosocial analysis : toward dialogues across disciplines", WIREs Water 4(2) 1–14. Hydrosocial research take the messiness of the socionatural world as a given and results in location-specific narrative case study analyses with limited or no attempt at generalisation. Socio-hydrology looks to generalise findings from
case studies through a system-approach using conceptual and mathematical models. "Socio-hydrological system" is thereby an abstract entity detached from the reality on the ground. We propose "socio-hydrological space" as a tool that helps to make the necessary intermediary step between the messy reality of the specific location (space) and the abstract system of conceptual and mathematical models. The primary function of SHS is as a lens through which to view the complex reality of specific cases in order to find patterns in human-river interactions, which can then be compared to patterns in other locations to see if further generalisation towards universal models is possible. Its use invites the researcher to have an open mind to the existence of expected or unexpected patterns in location-specific data using a thorough understanding of the location: society, economics, natural system, technical interventions, etc. Subsequently, other cases may be analysed in order to explore whether the same or different patterns occur. These patterns can then be generalised through the more formal conceptualisation of socio-hydrological systems. On the one hand SHS thereby relates to a specific space, on the other hand it helps to find general patterns of human-river interactions: is serves as a methodological intermediary step or bridge between hydrosocial research and socio-hydrology. The importance of such an intermediary step is illustrated by the differences between our findings on human-river relations in the Jamuna floodplain and those by Di Baldassarre et al. published in several papers for the Po valley. From Di Baldassarre et al.'s analysis of human-river relations in the Po valley it appears that two alternative responses exist in time and space (levees or adaptation). This same pattern would also be broadly recognisable in other high income countries where control of the river is a financial and technical possibility, such as The Netherlands (levees) or USA (some locations have levees, at others adaptation is required). However, society along the Jamuna show both responses at the same time in one region, but at different locations (SHS1 and SHS2), with a third intermediary response (SHS3). We speculate that the greater variety in Bangladesh is due to less government budget and more difficult technical circumstances (the Jamuna is of a scale that renders most civil engineering works unsuccessful), but this remains for now an unexamined suggestion.

[Figure]

If Di Baldassarre's findings are therefore taken to derive a general conceptual model for socio-hydrological systems along rivers, as in his subsequent publications with co-workers, the resulting models may be applicable to other rivers in similar conditions, but not to the Jamuna floodplain. Distinguishing socio-hydrological spaces in the field is therefore an important step in the search for generalisation of human-river interactions as they combine a place-based analysis with a presumption of the existence of general-isable patterns, without assuming that these patterns will be the same across the world. The proposition of using SHS to examine field data thereby also helps to overcome a bias towards high income, moderate climate regions in the study of (socio-) hydrology that was identified by James Linton (2008) in "Is the Hydrologic Cycle Sustainable? A Historical-Geographical Critique of a Modern Concept". Annals of the Association of American Geographers 98(3) 630-649. Boundaries around a system are always arbi-trary and selected in an attempt to analyse and address specific research questions, and a system can be also nested within a higher level system. However, the field data do suggest some boundaries as more logical or useful. In our case, the number of SHS that we found (three) is in first instance a result of the scale at which we explored the Jamuna human-river interactions (i.e. it is a result of the research scope/funding, not of the research question). However, we observe that the same pattern occurs along most of the Jamuna going downstream, until physical circumstances change too much and the river becomes tidal and under influence of cyclones. Going upstream, too, the pattern continues into India. While the three SHS we found are therefore first of all based on patterns in location-specific data, they can be generalised and used as a typology that can be applied elsewhere – but like the Po SHS they cannot be applied everywhere. It remains to be seen whether the same pattern of these three SHS oc-curs along other rivers and in other socio-economic conditions. However, patterns of SHS (such as the two options proposed by Di Baldassarre, or our three SHS) can be used to compare two different regions, as suggested by the reviewer. We could then find some regions where the options are similar to the Po valley, and other where they are similar to those in the Jamuna floodplain. And we think other patterns will exist. We

contend that these patterns do not constitute (formal, mathematically conceptualised) systems, but this may be an matter of vocabulary only. Thank you for pointing out the publications by Mostert (2017) and Hazarika et al. (2015). We will make sure to discuss these in our revised manuscript. Their work is indeed closely related to ours. However, as discussed above with SHS we are proposing a tool to rise above single cases studies, to help the comparison across cases. SHS offers the bridge between a specific case study, to identify patterns that can be compared to cases elsewhere.

Referee comment: On page 2 in line 24-26 the authors state that "interactions and feedback mechanism between hydrological and social processes in floodplains remain largely unexplored and poorly understood" citing Di Baldassarre et al. 2013a. However, since this paper in 2013 there have actually been quite some studies that have explored these interactions (just a few examples: Viglione et al. 2014, Chen et al. 2016, Ciullo et al. 2017, etc.) and in fact the authors do acknowledge this later in the manuscript (page 3, line 5-6).

Response: We will rephrase out statement, as it is indeed a little out of date. We will make sure to discuss Viglione et al. (2014), Chen et al. (2016) and Ciullo et al. (2017) in our revised manuscript.

Referee comment: On page 2 the authors state that there are currently two approaches to sociohydrology: qualitative studies and conceptual mathematical modelling studies. As I mention above, there are in fact other approaches (e.g. Mostert et al. 2017 and Hazarika et al. 2016) very similar to the approach that is presented here as a new approach.

Response: We will make sure to discuss Mostert et al. (2017) and Hazarika et al (2016) in our revised manuscript (see also our reply to the first comment).

Referee comment: The authors repeatedly state that running a conceptual mathematical model based on differential equations is much more data-demanding than the approach taken here. However, running a conceptual model like that does not require any data at all! Unless one wants to compare the model with real data, which would indeed make it more data-demanding, but I would argue that it would be just as data-demanding as the approach taken here. In fact, in my opinion, using surveys and interview data is a very data-demanding approach (although a very valuable and useful approach).

Response: We agree with the reviewer and will phrase our assessment more carefully in the revised paper, also taking into account his/her previous remarks on other socio-hydrology studies.

Referee comment: In the discussion the authors state that the division into SHS and the testing is an iterative process. From the descriptions it seems that the "low char" and the "high char" are quite different from each other, so I wonder why the authors did not update their SHS based on the analysis?

Response: We agree that low char and high char are quite different in elevation and population density because of their ages. As with the other SHS that we distinguish, we are here looking at general patterns of human-river interactions as compared to the other SHS. In comparison to SHS1 and SHS 3, both types of char are located in the river bed and their inhabitants are exposed to more regular flooding and river erosion (as the bar diagrams show). If we zoom in further using our extensive data set, we expect to be able to distinguish sub-SHS within the current three SHS. This is indeed what we are planning to do in the near future for SHS1 and SHS3. We will be publishing further research that finds differences in human-river interactions within SHS1, depending on the level of protection offered by the levee and the degree of urbanisation. These differences could be argued to constitute different SHS, but here it is the research objective that indeed determines whether further splitting up of one SHS is useful. So the scale of analysis to some extent determines the level of detail included in the SHS that are recognised, but not absolutely since merging the three SHS we distinguish does not make sense. This does not undermine the usefulness of looking for SHS in a specific area, it merely shows that it depends on the scale of
investigation which differences can be distinguished.

Referee comment: In the discussion the authors state that: "Each SHS shows distinct features when comparing flood-society interactions, proving that the dynamic interactions of floods is dependent on different hydrological and societal characteristics along the Jamuna River." The authors do indeed describe the different hydrological and societal characteristics of the three SHS, however, I miss the translation to the different dynamic interactions that follow from these characteristics. The description stops at describing the characteristics and does not describe the interactions and feedbacks that we are interested in in socio-hydrology. Are there in fact different ways of coping with floods in these three SHS? And if so, why do they behave differently? Which societal and hydrological combinations of characteristics lead to which kind of interactions?

Response: In the paper we described the different hydrological and societal characteristics of the three SHS as a narrative in which we showed how different flooding events co-shape societal behaviour (migration, livelihood changes, homestead type, etc.) of the SHS. Inhabitants of these areas have adapted to these different physical conditions. Each SHS therefore has different coping strategies with flooding. We will expand our discussion of this issue in the revised paper, as suggested by the reviewer.

Referee comment: In the conclusion, the authors conclude that the concept draws attention to how historical patterns of coevolution of social behavior, natural processes and technological adoptions give rise to different landscapes, different styles of living, and different ways of organizing livelihoods, while in fact the concept as it is presented here and applied to the case study, does not do this at all. It leaves me wondering what the different patterns, different styles of living, etc. are that emerged in these three SHS.

Response: We are a little puzzled by this comment. We think that our narrative description of the three spaces together with the statistical analyses of survey data show that three distinct socio-hydrological spaces with different landscapes, different styles

of living, and different ways of organizing livelihoods. Since we have not convinced the reviewer, we will expand the descriptive section accordingly.

Referee comment: A large part of the discussion is about the spatial boundaries. The authors stress the point that the boundaries of the SHS move in time and that the physical boundaries between the three SHS are not fixed in time. While this is true, I do not really see why this is of importance. The SHS you define are defined by the characteristics of the system, not by the exact coordinates. For example, the authors define SHS 2 as a char within the river, if the river moves a kilometer and the char moves with it (or a different char forms), this does not change the definition of SHS 2 as a char within the river. The same holds for the social boundaries, if one person moves to another SHS and adopts the strategies of that SHS, then the SHS does not change, does it? I think the authors could spend less attention on this in the discussion.

Response: We agree fully with the reviewer that the exact boundaries are not important for the use of SHS. Interestingly, reviewer 3 voiced exactly opposite concerns, namely that the exact boundaries are important. In anticipation of this second concern, we added this discussion to the paper. We will make this clearer in the revised paper. The boundaries are not fixed in time because the Jamuna River is very dynamic. The SHS definition will not change if the boundary changes over time but for this research we collected and analysed primary and secondary data based on 2016 bankline of Jamuna River. If the boundary changes and respondents move from one space to another space then it might affect the results we showed in this paper.

Referee comment: Figure 4 is not really consistent. The legend is placed in different locations, some graphs do show the total percentage on top of the bars and others don't (and some do but miss the %). Also, when printed in black and white, the difference between the color of SHS 1 and SHS 3 is not clear.

Response: We will replace figure 4 with an update version in the revised manuscript.

Referee comment: The format of figure 5 does not really allow for an easy comparison
between the three SHS, I would suggest choosing another type of figure.

Response: We will try to find another type of figure to show easy comparison. If possible we will replace it in our revised manuscript.

———————————————————

---

## Short Comment (SC5) · 28 Feb 2018

Replies to editor comment

We want to thank the editor for encouraging us to reply to the reviewers' comments. To a large extent, they are sympathetic to our proposal, but they also concur in stating that our proposed 'new approach' of using the concept of 'socio-hydrological spaces' to do socio-hydrology research is not yet well enough defined, its empirical use at other sites it not yet explained well enough, and the exact purpose of using socio-hydrological spaces is not yet argued well enough. We are grateful for these comments, because they help us to advance our own reflection about these issues. Our 'intellectual intuition' tells us that socio-hydrological spaces are a useful addition to the methods used in

socio-hydrology. This assessment is based on the one hand on our use of the concept to understand the socio-hydrology of the Jamuna flood plain. On the other hand, our proposition is based in our extensive exposure to different kinds of research, spanning the full range from deterministic modelling to narrative case studies, and our wish to bridge the (inter)disciplinary 'gaps' that result from these different approaches. We agreed that we need to 'sell' our proposition better, and the reviewers gave given us clues as to how we can do this.

Our overall argument runs as follows (which is repeated in our replies to individual reviewers). We think SHS provides a methodological (and possibly paradigmatic) bridge between two contrasting approaches to studying human-water interactions: hydrosocial research based in sociology and human geography, and socio-hydrology based in hydrology and physical geography. These are described and discussed in Wesselink, A., Kooy, M. and Warner, J. (2017) "Socio-hydrology and hydrosocial analysis : toward dialogues across disciplines", WIREs Water 4(2) 1–14. Hydrosocial research take the messiness of the socionatural world as a given and results in location-specific narrative case study analyses with limited or no attempt at generalisation. Socio-hydrology looks to generalise findings from case studies through a system-approach using conceptual and mathematical models. "Socio-hydrological system" is thereby an abstract entity detached from the reality on the ground.

We propose "socio-hydrological space" as a tool that helps to make the necessary intermediary step between the messy reality of the specific location (space) and the abstract system of conceptual and mathematical models. The primary function of SHS is as a lens through which to view the complex reality of specific cases in order to find patterns in human-river interactions, which can then be compared to patterns in other locations to see if further generalisation towards universal models is possible. Its use invites the researcher to have an open mind to the existence of expected or unexpected patterns in location-specific data using a thorough understanding of the location: society, economics, natural system, technical interventions, etc. Subsequently,

other cases may be analysed in order to explore whether the same or different patterns occur. These patterns can then be generalised through the more formal conceptualisation of socio-hydrological systems. On the one hand SHS thereby relates to a specific space, on the other hand it helps to find general patterns of human-river interactions: it serves as a methodological intermediary step or bridge between hydrosocial research and socio-hydrology.

To use the concept of SHS empirically, we propose a two-step approach. First, a thorough understanding of a specific floodplain system (geography, history, technology, societal occupation etc.) results in a preliminary classification of the study area into distinct SHS. Second, the classification is tested for statistical significance using available or newly collected data. If the classification is not statistically significant, merging or splitting of categories should be considered where different social dynamics may be the reason for splitting (repeat step 1). The concept suggests that the interactions between society and water are place bound because of differences in social processes and river dynamics, but also generalisable since similar SHS patterns may be found elsewhere. Rather than a generalized model for understanding how such interactions occur, the concept draws analytical attention to how flood dynamics co-evolve with societal dynamics. Such attention is useful anywhere in the world and for other socio-hydrological systems than floodplains.

We believe we can revise the paper using the above line of argument to more convincingly show that socio-hydrological spaces is a useful tool in socio-hydrology. We will also make sure to pay attention to the detailed comments and suggestions made by the reviewers.

---

## Editor Comment (EC2) · M. Sivapalan (Editor) · 28 Feb 2018

I thank the authors for their detailed responses to the reviewers. It seems like a reasonable response to me, but hopefully one or more of the reviewers will respond with additional comments or suggestions. While we await this feedback, I will also go over these issues and will make my recommendation about how to proceed. Please go ahead with your plans to revise the manuscript even while you wait for feedback from the reviewers and myself.

---

## Editor Comment (EC3) · M. Sivapalan (Editor) · 11 Mar 2018

I thank the authors for posting their responses to the four reviewers.

It seems that all four reviewers have raised questions about both the meaning and need for the introduction of the notion "socio-hydrological spaces". The authors have tried to explain it on the basis of their perceived need to build a bridge between socio-hydrology and hydrosocial research.

I am not 100% convinced, and I would like this to be resolved or the arguments improved through the review process. I am going to ask the authors to make the best case possible for introducing the notion and I am going to ask the reviewers to weigh in on this argument. For the future of socio-hydrology and for the validity of this paper,

this needs to be resolved, and resolved through the review process.

The responsibility is on the authors to make a convincing case in a revised manuscript that addresses not only this issue, but also other critical issues raised by the reviewers.

I look forward to the revised manuscript, and also look forward to being educated through the review process.

Note that in the second round of reviews will be not be public.

---

## Author Comment (AC1) · 19 Mar 2018

We have submitted our response and do not want to change this at this point.
* * *

---

## Author Comment (AC3) · 19 Mar 2018

We have submitted our response and do not want to change this at this point.

---

## Author Comment (AC4) · 19 Mar 2018

We have submitted out response and do not want to change this at this point.

---

## Author Response (AR1)

**HESS-2017-748 Socio-hydrological spaces in the Jamuna River floodplain in Bangladesh**

**Referees comments, Author's response and Author's changes in manuscript**

**1 Comments from referees/public**

**Comments from Anonymous Referee #1**

This is an interesting work based on a strong empirical and field based work. While I enjoyed reading the work, I was bothered by the concept of "socio-hydrological space" that the authors are pushing for. Why not just call it "social-hydrological system"? By calling it a "socio-hydrological space," what new things become possible that couldn't be achieved when you just simply call it socio-hydrological system? The notion of system has been around for long and it is exactly what the authors are trying to do. A system refers to an "integrated whole" and is composed of several interacting parts or 10 elements. Of course, this assumes the presence of a boundary delineating which parts are inside the system and which are outside of it. And this boundary can be of different forms: spatial boundary, organizational boundary, ecological boundary, you name it. People specify these boundaries in an attempt to analyze and address specific research questions. So, system boundary is arbitrary and a system can be also nested within a higher level system. Let me challenge the authors. Can you define a larger socio-hydrological space that includes those three socio-hydrological spaces you described in the paper? I'm 15 sure you could if you're comparing larger-level spaces between two very different regions. So, why not just use the term system? In social ecology, they use the term "social-ecological system." They don't use "social-ecological space."

I also would like to see more discussion on how flood coping strategies vary by SHS1-SHS3. The authors do describe something, but not detailed enough. More details on how individual level strategies (cropping pattern, migration strategies, home floodproofing) and group-level strategies (activities organized by communities) should be provided.

Figure 2 needs some improvement. Hard to see dotted line (levee). Hard to see boundaries of SHS1-3. If printed in B&W, these can't be distinguished.

I am also bothered by expressions like "adaptation space" and "levee effect space" in page 4. Adaptation and levee effect are emergent phenomena generated by system dynamics. I don't know what you mean by these can be rendered in terms of SHS. Quite a few awkward grammars here and there. E.g., "channels more and more move into" (page 8).

In page 15, the authors say "the concept provides a methodological and theoretical advance in the socio-hydrology." I am not convinced why this is so.

**Comments from Anonymous Referee #2**

Ferdous and colleagues developed a new concept called 'socio-hydrological spaces' which they define as a geographical area 30 with distinct hydrological and social features that give rise to distinct patterns and emergent behavior. They then apply this concept to an analysis of the Jamuna River floodplain in Bangladesh. In case study they identify three distinct socio-hydrological spaces defined by geographical features and support this delineation with primary and secondary data. The example application is well supported by primary data collection. The application of mixed-method approaches is important in socio-hydrology and the topic is of interest to HESS readers. However, I do have a series of concerns that if addressed would strengthen the paper. I believe that with certain revisions it would be suitable for publication.

Comments

1. The definition of 'socio-hydrological spaces' hints at two different types of spaces. The first is space as a geographical area. The second is space as a portion of the parameter space which leads to a distinct set of emergent dynamics. (The examples of the adaptation space and levy effect space on page 4 further raise the question of the second type of space.) In the case presented, geographical features (e.g. embankment) are used to divide the case area into three sub-areas with different dynamics. Because these geographic features define the dynamics of the system all of the unions exhibiting similar dynamics are spatially clustered. However, I can envision cases in which the features defining the socio- hydrological dynamics are social not physical features. In these cases, I am not sure the 'spaces' would be contiguous. How would this approach be applied to a case where geographical features are poorly aligned with system dynamics? Or is this tool suitable for only the cases where geographical features are aligned with system dynamics?

2. In the definition section (pages 3-4), the authors present this concept/tool as an alternative to either narratives or mathematical models. However, in the case that follows the authors present both the 'socio-hydrological space' delineation with a case narrative, which I think was effective. Rather than serving as an effective standalone tool, 'socio-hydrological spaces' compliments these other approaches. I think the author's argument for this tool would be more convincing if they could frame it as part of a broad research plan. For example, the authors note that SHS is descriptive not explanatory. If combined with other approaches could it enhance the explanatory power of a study?

3. While it is important to expand the approaches used to address socio-hydrological questions and to synthesize quantitative and qualitative data, this is not the first study to do so. The authors should acknowledge other efforts in this space such as data-driven narratives (Treuer et al. 2017) and the pairing of statistical analysis and narratives (Hornberger et al. 2015), and articulate what 'socio-hydrological spaces' adds.

4. I think there is potential for this concept to be used comparatively across say multiple flood plain cases. Please speak to this potential.

5. Lastly, there are some typographic errors and awkward phrasing in the manuscript and it would benefit from a thorough review.

References

Hornberger GM, Hess DJ, Gilligan J (2015) Water conservation and hydrological transitions in cities in the United States. Water Resour Res 51(6):4635–4649.

Treuer G, et al. (2017) A narrative method for analyzing transitions in urban water management: The case of the Miami-Dade Water and Sewer Department. Water ResourRes 53(1):891–908.

**Comments from Anonymous Referee #3**

The authors aim to present a "new way of looking at and analysing socio-hydrological systems", and use a study area in the highly dynamic floodplains of the Jamuna river in Bangladesh.

After reading the introduction, I wanted to know: -how to construct and define a SHS -how the SHS improves or benefits the field of socio-hydrology, -how to apply the SHS to other research areas, or even to other areas within the country.

General comments

I am impressed by the authors' knowledge of the study area. The methods used to construct the survey seem sound, the questionnaires (ESM1) suitable to the research question at hand. The topic is of interest to HESS readers.

Overall, the paper seems to be a further development to the classification performed in Di Baldassarre 2013 and 2015 and rests on the assumption of two patterns of society river interactions. (see also p. 5, line 1-5). While I understand that the concept of SHS is new to the field in terms of vocabulary, I don't see why the classification from Di Baldassarre 2015 which is criticized by the authors cannot simply be performed on a smaller scale. Do we need SHS for that? How could the SHS concept be extended to the entire country? I am also not convinced by the results that this approach and the presented results draw "analytical attention to how flood dynamics co-evolve with societal dynamics".

My initial questions were only answered partly. I am unfortunately not sure how this method is an improvement or benefit to the field of socio-hydrology. I see the study's strong point in the extensive empirical field survey, but feel that this requires more work to show statistical relationships gathered from the individual SHS and then comparing those to hydrological data (flood extent, erosion, etc.). ). I also cannot easily detect how the SHS approach is useful in specifying the interaction between sociological and hydrological processes in the sense of the two-way feedbacks key to socio-hydrological approaches.

At present, SHS still seems to be a rather descriptive and classical approach to me, with the statistical methods mainly from the field of basic exploratory data analysis. While there is no harm in that, the authors do stress that they present a "new approach". The extensive surveys should be brought into context with actual observed data (especially Sec 5.1-5.4), in particular if the authors consider using this approach to make predictions (although it is unclear to me what they wish to predict and how, this is only mentioned in the beginning of the paper and should definitely be elaborated on). The authors should also address the uncertainties in their work – there are a lot of biases inherent in conducting surveys, and I'm not sure the time span 1960-2016 is feasible due to the large number of external factors that could also contribute to e.g. migration or farm land area (such as the independence of Bangladesh in 1971).

Minor and major remarks

Section 3.1: Not entirely sure why this specific study area was chosen. How big are the individual SHS? Section 3.2: Why are socio-economic factors not relevant for the construction of the SHS? I would assume this makes a difference in how the livelihoods are affected by flooding in the individual SHS. (Add-on: on p.8, line 16 the occupation is listed as a delineating factor. I don't see this in section 3.2, where the delineation is based on "differences in geophysical characteristics and flood protection measures". Section 3.3: Is "evidence'' the right word to use? Perhaps "data'' is more suitable. Sections 4.1-4.3:

can these be classified as Results? I would consider this to be part of the methodology/study area description. (VERY narrative). Section 4.2 Could benefit from references on chars. Sometimes, chars is in quotes, most of the time not. Please be consistent. Section 5.1 How was this verified? Using the household's answers can be deceptive, as there is a strong bias to the length of time since the last flood event. Also, of course a char in the Jamuna cannot be flooded by another river.

Sections 5.4 Using only household surveys to state that e.g. "riverbank erosion is experienced in each zone", and to comment on how high these rates are without presenting physical observations is in my opinion not conclusive. I strongly suggest backing these statements up with observed erosion data. Also, how far away from the river do your respondents live? This can bias the answers, making the statements even more inconclusive. Section 5.7: how is this section relevant?

p. 3, line 24ff You state that the concept's importance lies "in its emphasis on how the interactions between society and
water are always place-bound'' Perhaps I use a different interpretation for the word place-bound, but the levee effect you mention afterwards is anything but place-bound. Rather, you describe yourself how this was introduced for the Po floodplain as well as by White in the US. Please clarify. P.5, line 3ff please clarify what you mean in this sentence – unclear to me. p.5, line 17: I am surprised that you do not mention any of the extreme flooding after 2007 – just last year severe flooding in the region occurred. On p.6, line 10 you do mention the flood of 2016, so please check for consistency. Perhaps it would also be
good to just name those years in which the study area was extremely flooded, not "general'' extreme flood years in Bangladesh. p.6, line 11: how much percent was flooded? p.8, line 1: when were the focus group discussions with respect to the study years and the flood season? Also, during which season/months were the surveys conducted? p.8, line 10: Frequency analyses for what? The following sentence is unclear. p.8, line 26: how much of the bankline is eroded? p.9, line 29: migration to where? Outside of SHS3? p.10, line 17: is the unexpected flood frequency observed through e.g. data or
satellite imagery? p.11, line 6: please include the flood damage information in the description of data and methods. How did you analyse what? How do the individual floods compare with respect to magnitude and flood duration in the individual SHS in each of those years? What about the study years? p.12, line 1: is there a citation for this? How low is the average elevation? It would be good to include this in the general description of the SHS. p.12, line 13: how is the number of farmers with large households determined? If only from the questionnaire, how did you control for other biases such as migration,
change of occupation? How certain do you think this number is? I would argue that the changes in SHS1 are not significant, and that they in particular cannot be attributed solely to consecutive flood events. Also, why 1960? Does it not make it more difficult to evaluate the results before/after Bangladesh became independent? When was the embankment in SHS1 built? Could the reduction of large farms not simply be due to other socio-economic developments in the region? Is this also solely based on information from questionnaire? p.12, line 24: this is a major concern I also share in these types of studies (and I
am not convinced this can be verified through a focus group discussion- how?). This is also why I stress the need for observed data. p. 13, line 7: increased by how much over what time period? p. 13, line 15: Did the respondents arrive or leave? The last two sentences of this paragraph are unclear to me. p. 14, line 11: which interactions between sociological and hydrological processes did you identify? Which two-way feedback are you referring to? p. 14, line 20: when do you consider the initial selection of the SHS to have "statistical meaning"? How is this transferable? How can you be sure they are consistent over time? What is the added value of SHS if their boundaries are mobile? p. 15, line 18 ff: I agree. Please expand your methodology to include when your selected SHS need to be updated – for now, this is not quite clear. p. 15, line 22: what advance did you show? Which questions did you now answer that could not be answered before? How can you apply this in a broader sense? p. 16, line 16: To which policies, for example? What is a "rapid rural appraisal"?

Literature

Literature cited: the work largely cites and even uses figures from the same two papers (Di Baldassarre 2013a and b). While this is of course expected when developing the work of one research group further, what exactly is the point referencing literature such as the authors did in p.3, line 14 or p.4, line 15? I suggest to simply let the reader know where to look for the information or statement in the sentence before. FICHTER and nhc, 2015 does not look correct.

Language

Language is mostly good, but could definitely benefit from a careful read-through by a native speaker or a language editing service. E.g., p 2 Line 30 sees three uses of the word "different'' in one sentence, and there are numerous grammatical or typographic mistakes throughout the paper. Be careful to introduce abbreviations before you use them (e.g. in abstract). p.5, line 27: "To evidence and understand: " sounds a bit awkward p.8, line 21: "inundated" instead of "ponded"? p. 15, line 14:

what is "people mobility"?

Figures

Figure 2: cannot decipher the black names when printing out copy, perhaps resolution needs to be better. I had to look really closely to detect the boundaries of the individual SHS. Why is the land colored red? Figure 4: please include the number of respondents for each subset. Figure 5: Perhaps consider labeling all axes outside of plot or all inside plot (consistency). Also, it should say "SHS3". Starting when can a farmer be considered to sustain the own household? Figure 7: why is the land red? Why was the dry season chosen? A different coloring would greatly benefit the readability of the figure.

**Comments from Anonymous Referee #4**

The authors propose a new concept to study the interactions between humans and floods in a socio-hydrological system.

They introduce the concept of Socio- Hydrological Spaces to describe a system that shows specific interactions between social, economic, hydrological, etc. factors that result in a certain behaviour of the system and apply this to a case study in Bangladesh.

Although I can understand the advantages and potential of a comprehensive systematic approach to the study of "Socio-Hydrological Spaces" (which the authors seem to be aiming at) this new approach is quite poorly defined and explained. The authors merely give human-flood systems a different name (i.e. Socio-Hydrological spaces) and proceed to describe a case study as if this is a new approach. Mostert (2017) recently published an article in this same journal, arguing for case-study research as an alternative approach for socio-hydrology and while his example of a case study is perhaps more qualitative than the one presented here, the authors should perhaps try to relate to his paper. Also, a very similar approach to the one presented in this manuscript for describing a case study of how humans and floods coexist, is presented by Hazarika et al. (2015).

The concept/approach would be new and in my opinion useful, if a general framework would be presented to analyze a case study/SHS in a comprehensive and consistent way, which would allow for the comparison of different Socio-Hydrological Spaces, their specific characteristics, and the feedbacks and phenomena that arise from the characteristics of this particular system. However, after reading the manuscript I did not really see how the method/concept that is presented here adds something new and useful to the already existing approach of a case study description.

Some more specific comments:

1) On page 2 in line 24-26 the authors state that "interactions and feedback mechanism between hydrological and social processes in floodplains remain largely unexplored and poorly understood" citing Di Baldassarre et al. 2013a. However, since this paper in 2013 there have actually been quite some studies that have explored these interactions (just a few examples: Viglione et al. 2014, Chen et al. 2016, Ciullo et al. 2017, etc.) and in fact the authors do acknowledge this later in the manuscript (page 3, line 5-6).

2) On page 2 the authors state that there are currently two approaches to sociohydrology: qualitative studies and conceptual mathematical modelling studies. As I mention above, there are in fact other approaches (e.g. Mostert et al. 2017 and Hazarika et al. 2016) very similar to the approach that is presented here as a new approach.

3) The authors repeatedly state that running a conceptual mathematical model based on differential equations is much more data-demanding than the approach taken here. However, running a conceptual model like that does not require any data at all! Unless one wants to compare the model with real data, which would indeed make it more data-demanding, but I would argue that it would be just as data-demanding as the approach taken here. In fact, in my opinion, using surveys and interview data is a very data-demanding approach (although a very valuable and useful approach).

4) In the discussion the authors state that the division into SHS and the testing is an iterative process. From the descriptions it seems that the "low char" and the "high char" are quite different from each other, so I wonder why the authors did not update their SHS based on the analysis?

5) In the discussion the authors state that: "Each SHS shows distinct features when comparing flood-society interactions, proving that the dynamic interactions of floods is dependent on different hydrological and societal characteristics along the Jamuna River." The authors do indeed describe the different hydrological and societal characteristics of the three SHS, however, I miss the translation to the different dynamic interactions that follow from these characteristics. The description stops at describing the characteristics and does not describe the interactions and feedbacks that we are interested in in socio-hydrology. Are there in fact different ways of coping with floods in these three SHS? And if so, why do they behave differently? Which societal and hydrological combinations of characteristics lead to which kind of interactions?

In the conclusion, the authors conclude that the concept draws attention to how historical patterns of coevolution of social behavior, natural processes and technological adoptions give rise to different landscapes, different styles of living, and different ways of organizing livelihoods, while in fact the concept as it is presented here and applied to the case study, does not do this at all. It leaves me wondering what the different patterns, different styles of living, etc. are that emerged in these three SHS.

6) A large part of the discussion is about the spatial boundaries. The authors stress the point that the boundaries of the SHS move in time and that the physical boundaries between the three SHS are not fixed in time. While this is true, I do not really see why this is of importance. The SHS you define are defined by the characteristics of the system, not by the exact coordinates. For example, the authors define SHS 2 as a char within the river, if the river moves a kilometer and the char moves with it (or a different char forms), this does not change the definition of SHS 2 as a char within the river. The same holds for the social boundaries, if one person moves to another SHS and adopts the strategies of that SHS, then the SHS does not change, does it? I think the authors could spend less attention on this in the discussion.

7) Figure 4 is not really consistent. The legend is placed in different locations, some graphs do show the total percentage on top of the bars and others don't (and some do but miss the %). Also, when printed in black and white, the difference between the color of SHS 1 and SHS 3 is not clear.

8) The format of figure 5 does not really allow for an easy comparison between the three SHS, I would suggest choosing another type of figure.

**2 Author's final response to referees**

Dear Professor Sivapalan,

We have submitted a revised version of our paper "Socio-hydrological spaces in the Jamuna River floodplain in Bangladesh". Following the reviews, we have improved and expanded our sections on the reasons for and definition of socio-hydrological spaces. We have also fixed the more minor issues raised by reviewers. Below, we respond to the main reviewers' comments.

We propose the concept of socio-hydrological spaces (SHS) as a useful 'tool' in the research on socio-hydrology of floodplains (and probably elsewhere). The critique by the reviewers on this proposition can be summarised as three questions:

1. What is a SHS?

2. How do you apply SHS to a case?

3. Why should we have SHS in addition to existing concepts = what use is SHS for socio-hydrology?

The second question has answered by adding more detail to the paper. The first and last questions are more difficult because they relate to the paradigm(s) in use in socio-hydrology. In summary, what we do try to do is to bridge the gap between modelling and rich case descriptions of reality. We have now expanded the paragraphs in the Introduction that explaining this, and added a separate section 'Patterns in hydrology' to provide a better underpinning of our proposal. With the proposal of SHS we are in essence saying that we are looking for a middle ground where we preserve the variability of reality and the unpredictability of human behaviour and decisions, not fit these into a model (which could be "there is either fight or adapt"), while at the same time recognising patterns (combinations of the same fight and/or adapt responses).

This middle ground is reflected in the expanded definition of SHS in Section 3, and below:

*Definition of SHS: bottom up (data driven) vs top down (hypothesis/model driven)*

In the paper we define a SHS as "a geographic area with distinct hydrological and social features that gives rise to the emergence of distinct interactions and dynamics between society and water, which also vary with time a space". This can be considered a data driven definition. We currently label these distinct interactions and dynamics as 'levee effect' and adaptation effect'. I believe we should re-label these 'fight' vs 'adapt' since the levee effect is a result of fighting, not action itself.

SHS can also be defined from the general patterns found in human-flood interactions (as in Di Baldassarre et al.) as "a geographical area where the empirical/concrete manifestation/expression of a specific combination of generic socio-hydrological patterns (here: fighting and adaptation dynamics) is distinct from neighbouring areas". This can be considered a model driven definition. This top down definition has been added to the paper.

Both definitions apply simultaneously, and are used in iterative mode to study the socio-hydrology of an area. The top down definition guides the researcher to look for these generic patterns in the area of study, resulting in preliminary delineation of SHS and their qualitative descriptions (or narratives); these results have the function of being hypotheses in the next step. In this next step, spatial data analysis to confirm/reject these initial hypotheses, that is, they provide the data driven (bottom up) delineation of SHS.

SHS is thereby a missing link between a specific (concrete) case (e.g. Jamuna or Po floodplain) and generic (abstract) patterns = a link between observations and theory/model. The concept of SHS is needed because it turns out that the generic patterns that seem to map 1-to-1 onto reality in certain cases (a whole floodplain (in time or space) or when applied at river basin scale) do not map 1-to-1 onto reality in Bangladesh, or when floodplains are examined in detail. Hence: SHS are needed to refine the analysis of human-flood interactions from the generic (either fight or adapt) to the concrete and local (where both may co-exist in time). SHS allow better representation of the reality of these interactions, while at the same time enabling comparison between cases by referring to generic patterns.

This is what we put in our earlier reply to reviewers, which summarises the above: "We propose "socio-hydrological space" as a tool that helps to make the necessary intermediary step between the messy reality of the specific location (space) and the abstract system of conceptual and mathematical models. The primary function of SHS is as a lens through which to view the complex reality of specific cases in order to find patterns in human-river interactions, which can then be compared to patterns in other locations to see if further generalisation towards universal models is possible. Its use invites the researcher to have an open mind to the existence of expected or unexpected patterns in location-specific data using a thorough understanding of the location: society, economics, natural system, technical interventions, etc. Subsequently, other cases may be analysed in order to explore whether the same or different patterns occur. These patterns can then be generalised through the more formal conceptualisation of socio-hydrological systems. On the one hand SHS thereby relates to a specific space, on the other hand it helps to find general patterns of human-river interactions."

**Reflection on other comments by the referees**

**Referee #1** asked why we do not use 'socio-hydrological system' instead of SH space, since system is an accepted and widely used concept (this implies: we should apply the principle of parsimony). (S)he also points out that a system can be delineated by other types of boundary than a spatial one (this implies: system is more widely useful as a concept). Our reasons are that:

- A system may be conceptual/abstract or concrete/real, which may lead to confusion between the two when used in a text;
we want to focus on the concrete and not leave any doubt about that. Worse, from this confusion, using 'system' may lead to the conflation of one with the other as if the conceptual system represents the reality (near) perfectly – which is the proposition made by many socio-hydrologists (see above).

- Because we want to start with a concrete situation to delineate SHS, we need concrete, visible and tangible boundaries = spatial ones.

**Referee #2** raises similar issues. (S)he points out that there is an ambivalence in the paper between SHS as a spatial entity and SHS as a portion of the parameter space that has distinct emergent properties. With the revised definition of SHS above, we make explicit that SHS has both characteristics. On the other hand, reality cannot be fitted completely into the parameter space of a model, and we do not want to limit our understanding of the SHS that we find in BD to this parameter space. Like
reviewer #1 (s)he suggests that the SHS could equally be delineated by social boundaries (instead of spatial ones), because the system's dynamics are not necessarily aligned with geographical features. While this is true in theory, in the practice of the study of human-flood interactions it is found that space constitutes the  boundaries within which humans operate and organise themselves to deal with floods.

**Referee #3** asks why we do not scale up and lump a larger area (such as the whole study area, or larger) into one SHS. Reference to our definition should answer this: in a larger area the characteristics are no longer the same and different from other areas. Reviewer #3 is not happy that we have provided statistical proof that the SHS are distinct. We should stress that we have used the ANOVA test to prove the classes are distinct for all graphs, and also that we can only illustrate the concept here and not provide all details. Related to this, (s)he finds the shifting spatial & societal boundaries (as discussed in our
Discussion) problematic. Interestingly reviewer #4 has no problem with this: "The SHS you define are defined by the characteristics of the system, not by the exact coordinates. For example, the authors define SHS 2 as a char within the river, if the river moves a kilometre and the char moves with it (or a different char forms), this does not change the definition of SHS 2 as a char within the river. The same holds for the social boundaries, if one person moves to another SHS and adopts the strategies of that SHS, then the SHS does not change." We have thanked reviewer #4 for this insight in a footnote.

To respond further to these three comments, we would like to add that the resulting ambivalence between SHS as a spatial entity and SHS as a portion of the parameter space that has distinct emergent properties is intentional: SHS has both characteristics. Reality cannot be fitted completely into the parameter space of a model, and we do not want to limit the understanding of the SHS that we find in reality to this parameter space. This is also one of the reasons why we use 'space'

instead of 'system'. A system in socio-hydrology may be conceptual/abstract or concrete/real, which may lead to confusion between the two when used in a text; we want to focus on the concrete and not leave any doubt about this, which the use of 'system' may do. Worse, from this confusion, using 'system' may lead to the conflation of one with the other, as if the conceptual system represents the reality (near) perfectly, which we do not believe is possible. Because we want to start with a concrete situation to delineate SHS, we need concrete, visible and tangible boundaries, which are spatial ones. The spatial boundaries of SHS may be physical, such as a river bank, but also due to social characteristics, such as a distinction between rural and urban land use which will give rise to other socio-hydrological dynamics. The other reason why we introduce the concept of SHS alongside 'socio-hydrological system' is related to complexity. 'System' is the appropriate term for an entire delta, which can be divided into subsystems depending on research goals. The smallest of these subdivisions would be a SHS, where socio-hydrological dynamics are unique. A similar distinction can be found in ecological systems research, where one ecosystem contains many habitats as smallest unit of study. We considered adding this paragraph to the paper, but decided it may detract readers by raising another issue, so omitted it in the revised submission.

**Referee #4** posits that the concept of SHS would be useful if it was a generic framework for analysing a case study. It could then be applied in a consistent and comprehensive way to enable comparison across cases. (Ss)he finds that the use(fulness)

of the SHS concept currently does not raise above the Jamuna case. Hopefully the new definition and better explanation will convince reviewer #4. (S)he also thinks that the current description of the three SHS only lists human and flood characteristics separately and omits the interactions. If we agree that the two responses 'fight' and adapt' present the two principal types of interaction, we can use these to label the three SHS more explicitly with 'fight' and/or 'adapt' to include the respective interactions are requested.

We hope you consider this manuscript to be ready for publication in HESS.

Kind regards,

Ruknul Ferdous and co-authors.

**Socio-hydrological spaces in the Jamuna River floodplain in Bangladesh**

[revised manuscript text omitted]
 mostlyrelocated within the study area knew that their new places aredestination was flood prone and also experienceat risk from riverbank erosion. However, the lack of available land is a major problem and the reason whyso they move to a risk prone areacontend with sub-optimal conditions.

**5̶6̶.7 Homestead types**

The construction type of the houses is different between the spaces (Fig. 6). Most of the pucca houses (well-constructed buildings using modern masonry materials) and semi-pucca or half pucca houses (made of brick and tin) are within the SHS1

and SHS3, where people feel comparatively safe against flooding and erosion. As a result, they invest more in their home. In SHS2 a high proportion of kutcha (wood, straw and bamboo mats) and jhupri (straw) houses is observed, since these are easy to take apart and move in case of flooding or erosion, and less costly to construct.

[Figure]

[Figure]

**Figure 6: Homestead type of households.**

**7 Discussion**

~~We introduced the concept of socio-hydrological spaces (SHS) and applied it to a test area along the Jamuna River in Bangladesh. We found it convenient and useful for categorizing and specifying the interaction between sociological and hydrological processes in the three identified SHS we distinguished in this location. The concept draws attention to the historical patterns of the hydrological processes of the Jamuna River and as well as different social processes along the three~~

Based on thorough in-depth knowledge of the natural, technical and social conditions of the study area in the floodplain of the Jamuna River in Bangladesh, we proposed distinguishing between three SHS as the basic spatial units each with distinct socio-hydrological characteristics. Human-flood dynamics are different in each space, ranking from 'adapt to floods' (SHS3), to more (SHS1) or less (SHS2) 'fighting floods' in combination with 'adapt to floods' to the extent necessary. We then proceeded to demonstrate, through statistical analysis of primary and secondary data, that the SHS show significant differences in the following hydrological and social variables: perceptions of the sources of flooding, flood frequency, flood damages, average household income and wealth, river bank erosion, migration, and homestead types. Our goal was not to prove that these boundaries to the SHS are the best, since this would require much more detailed social and hydrological data than currently available. However, we did show that there are good reasons to consider the three SHS as distinct both from a narrative and from a statistical perspective, and that such a distinction provides a good starting point for further socio-hydrological analysis of human-flood dynamics of the area.

The issue of drawing boundaries around the SHS merits some further discussion. We started by outlining the boundaries of three SHS based on the presence of distinct physical features in the landscape: the embankment on the west bank, the natural levee on the east bank, and the riverbed in between.  The  exact boundaries were drawn on pragmatic grounds, using the administrative boundaries that best align with the physical features. These boundaries might show  SHS in the present, but  the  boundaries of the physical and social systems are not fixed in time. The physical boundaries of the SHS are quite dynamic due to continuing bank erosion along both banks of the Jamuna (CEGIS, 2007). In particular, by analysing satellite images of the case study area from the late 1960s up to now, it appears that the west bank has been migrating westward and the east bank has been migrating eastward. As a result, the length-averaged width of the river has increased from 8.17 km to 11.68 km (CEGIS, 2007). Since the construction of the BRE in the 1960s, many breaches have occurred due to river bank erosion, forcing relocation of the embankment in many places (RBIP, 2015). At the same time due, to erosion of the east bank the natural levee also moved somewhat over time. Thus the physical boundaries between SHS1-SHS2 and SHS2-SHS3 are not fixed in time,

[Figure]

**while our statistical analyses assume**

 that they are since they use the current physical boundaries. The social boundaries of the SHS  in the Jamuna floodplain are also dynamic. Due to frequent relocations and migrations, the current inhabitants of the SHS may not have lived there throughout the study period since in every extreme event some migration occurs among the spaces (see Section 6). Therefore, the social boundaries between SHS1-SHS2 and SHS2-SHS3 are not fixed in time either, while our statistical analyses assume that they are since they relate to the current inhabitants. The dynamic nature of the boundaries of the SHS is unavoidable and indeed intrinsic to the highly dynamic socio-hydrology of the floodplain system. It is therefore important to remember that the SHS are defined by their unique socio- hydrological characteristics compared with the surrounding area, not by their exact coordinates. For example, SHS2 is defined as a char within the river. If the river moves a kilometre and the char moves with it (or a different char forms), this does not change the definition of SHS2 as a char within the river. The same holds for the social boundaries, if one person moves to another SHS and adopts the strategies of that SHS, then the SHS does not change[1]. Ideally, the data collection and analyses of time series in Step 2 would follow these shifting boundaries, but this will most likely not be possible for due to data scarcity or time constraints.
* * *
[1] We would like to thank one of the anonymous reviewers for helping us to formulate this insight.

[Figure]

**Figure 7: Time series dry season satellite images of the case study area.**

We tested the SHS concept in a quick changing socio physical environment, namely a floodplain in Bangladesh, and find that the concept provides a methodological and theoretical advance in the socio hydrology of floodplains as it helps identifying and categorizing human-water dynamics in specific geographical locations. We believe that the concept has a broader validity and can be applied to identify micro socio-hydrological contexts in other floodplains, characterized by different socio physical features.

Finally, a step forward in this research topic is the application of the SHS methodology shown here for the Jamuna floodplain to analyse physical processes other than floods, such as drought, salt intrusion, irrigated catchments or urban systems.

Conclusions

Socio-hydrological space (SHS) is a concept that enriches the study of socio-hydrology because it helps understand the detailed human-water8 Conclusions

We introduced the concept of socio-hydrological spaces (SHS) and applied it to a floodplain area along the Jamuna River in Bangladesh. SHS delineate areas where the interaction between social and hydrological processes show distinct characteristics, which in the case of floodplains can be expressed as different combinations of two basic patterns already identified in the literature: 'fight floods' or 'adapt to floods'. Applying SHS to other floodplains will enable global comparison of the existence and effects of these patterns in human-flood 
[revised manuscript text omitted]

---

## Referee Report (RR1)

The authors made efforts to improve the paper and explain their rationale behind the concept of socio-hydrological space (SHS). Thank you for that. However, despite the attempt, I still find their explanation of the concept and rationale behind it to be esoteric and verbose. If one cannot explain his/her ideas in a plain (clear & obvious) manner, it is probably a case of overdoing.

The authors say SHS is a middle ground between two methods: generic modeling and placed-based modeling (or specific case studies or attempts to exploring applicability of a placed-based model to other sites). It is also described as "the empirical expression of a specific combination of generic patterns (here: fighting and adaptation dynamics) in a geographical area that is distinct from the neighbouring one." I find these to be abstruse and difficult to understand.

Whether generic/abstract modeling or placed-based modeling, the approach is still the same for both—deductive modeling for various purposes (e.g., prediction, uncovering/testing hypotheses or underlying mechanisms that generate observed phenomena, explore the future possibility space, etc.) that are difficult to achieve by the means of other methods. Place-based modeling uses more specific assumptions and parameter values that are based on and motivated by a specific case/site. Generic models can be purely based on stylized facts/theories or motivated by certain recurring themes emerging from comparative analysis of multiple cases. A fundamental purpose of any modeling is not to capture reality as close as possible. Rather, models are like maps with only essential details. They are most useful when they contain only essential details. There is also bi-directional feedback between modeling and empirical studies. Empirical studies motivate modeling. Insights learned from modeling can be used to re-visit empirical case studies and refine knowledge gained. So, I don't understand the author's criticism about models "not being realistic" and it is also unclear to me what can be middle ground between the generic models and placed-based models. These are just different styles of modeling differentiated by modelers' intent or data availability.  The figure below is useful for thinking about the mutually reinforcing feedback between empirical studies and modeling studies.

[Figure]

Also, this expression in the paper "SHS is also the empirical expression of a specific combination of generic patterns (here: fighting and adaptation dynamics) in a geographical area that is distinct from the neighbouring one" is a good example of why I'm saying the authors' explanation is esoteric and verbose. I don't think readers who are not so familiar with sociohydrology will understand this. If one cannot explain in plain words, chances are that it is case of overdoing

So what can be done now? One way to salvage this work is taking a step back and re-think about how to simplify the story of the paper. The concept of SHS seems to emphasize rich historical patterns in a particular place. Thus, I think a good example to follow is Erik Mostert's recent paper in HESS (Mostert, E. (2017), An alternative approach for socio-hydrology: case study research, Hydrol. Earth Syst. Sci. Discuss., 1–14, doi:10.5194/hess-2017-299). The authors might reframe the paper as (1) emphasizing the importance of historical patterns in particular geographies for fully understanding generic as well as place-based models (show how) as well as (2) demonstrating how combining such rich historical patterns can help mutual reinforcing feedback between empirical studies and modeling studies.

---

## Author Response (AR2)

**HESS-2017-748 Socio-hydrological spaces in the Jamuna River floodplain in Bangladesh**

**Referees comments, Author's response and Author's changes in manuscript**

**1 Comments from editor, referees/public**

**Editor's comments and guidance:**

5 Editor Decision: Reconsider after major revisions (further review by editor and referees) (09 Jul 2018) by Murugesu Sivapalan

I have now received further comments from 3 of the original reviewers. They agree that the paper has seen considerable improvement, but they remain unsatisfied that the revised manuscript is in a form ready for publication. They provide considerable guidance towards possible areas that future improvements can be made.

10 I am now satisfied that a case now exists for this manuscript to be published but only one more round of revisions. Please study these comments carefully and come up with a plan/strategy of possible revisions that will satisfy the reviewers. I will be happy to provide feedback before you actually undertake the final revisions.

However, the final decision on any future manuscript will depend on the opinions of the reviewers. I will go for one more (hopefully final) round of reviews before I make the final decision. I promise to make the next round of reviews reasonably

15 fast so there is a chance for the paper to be published before the end of the year (at the latest).

Non-public comments to the Author:

I have ticked the major revisions box, I am hoping that it will not involve major new work. However, it is entirely up to you how you want to address the criticisms.

**Comments from Anonymous Referee #1**

20 Although the paper improved, I still find the authors' explanation of and rationale behind the concept of sociohydrological space to be esoteric and verbose.

The authors made efforts to improve the paper and explain their rationale behind the concept of socio-hydrological space (SHS). Thank you for that. However, despite the attempt, I still find their explanation of the concept and rationale behind it to be esoteric and verbose. If one cannot explain his/her ideas in a plain (clear & obvious) manner, it is probably a case of

25 overdoing.

The authors say SHS is a middle ground between two methods: generic modeling and placedbased modeling (or specific case studies or attempts to exploring applicability of a placed-based model to other sites). It is also described as "the empirical expression of a specific combination of generic patterns (here: fighting and adaptation dynamics) in a geographical area that is distinct from the neighbouring one." I find these to be abstruse and difficult to understand.

Whether generic/abstract modeling or placed-based modeling, the approach is still the same for both—deductive modeling for various purposes (e.g., prediction, uncovering/testing hypotheses or underlying mechanisms that generate observed phenomena, explore the future possibility space, etc.) that are difficult to achieve by the means of other methods. Place-based modelling uses more specific assumptions and parameter values that are based on and motivated by a specific case/site. Generic models can be purely based on stylized facts/theories or motivated by certain recurring themes emerging from comparative analysis of multiple cases. A fundamental purpose of any modeling is not to capture reality as close as possible. Rather, models are like maps with only essential details. They are most useful when they contain only essential details. There is also bi-directional feedback between modeling and empirical studies. Empirical studies motivate modeling. Insights learned from modeling can be used to re-visit empirical case studies and refine knowledge gained. So, I don't understand the author's criticism about models "not being realistic" and it is also unclear to me what can be middle ground between the generic models and placed-based models. These are just different styles of modeling differentiated by modelers' intent or data availability. The figure below is useful for thinking about the mutually reinforcing feedback between empirical studies and modelling studies.

[Figure]

Also, this expression in the paper "SHS is also the empirical expression of a specific combination of generic patterns (here: fighting and adaptation dynamics) in a geographical area that is distinct from the neighbouring one" is a good example of why I'm saying the authors' explanation is esoteric and verbose. I don't think readers who are not so familiar with sociohydrology will understand this. If one cannot explain in plain words, chances are that it is case of overdoing.

So what can be done now? One way to salvage this work is taking a step back and re-think about how to simplify the story of the paper. The concept of SHS seems to emphasize rich historical patterns in a particular place. Thus, I think a good example to follow is Erik Mostert's recent paper in HESS (Mostert, E. (2017), An alternative approach for socio-hydrology: case study research, Hydrol. Earth Syst. Sci. Discuss., 1–14, doi:10.5194/hess-2017-299). The authors might reframe the paper as (1) emphasizing the importance of historical patterns in particular geographies for fully understanding generic as well as place-based models (show how) as well as (2) demonstrating how combining such rich historical patterns can help mutual reinforcing feedback between empirical studies and modeling studies.

**Comments from Anonymous Referee #2**

Ferdous and colleagues developed a new concept called 'socio-hydrological spaces' which they define as a geographical area with distinct hydrological and social features that give rise to distinct patterns and emergent behavior. They then apply this concept to an analysis of the Jamuna River floodplain in Bangladesh. In case study they identify three distinct socio-

5  hydrological spaces defined by geographical features and support this delineation with primary and secondary data. The example application is well supported by primary data collection. The application of mixed-method approaches is important in socio-hydrology and the topic is of interest to HESS readers. However, I do have a series of concerns that if addressed would strengthen the paper. I believe that with certain revisions it would be suitable for publication.

Comments

10  1. The definition of 'socio-hydrological spaces' hints at two different types of spaces. The first is space as a geographical area. The second is space as a portion of the parameter space which leads to a distinct set of emergent dynamics. (The examples of the adaptation space and levy effect space on page 4 further raise the question of the second type of space.) In the case presented, geographical features (e.g. embankment) are used to divide the case area into three sub-areas with different dynamics. Because these geographic features define the dynamics of the system all of the unions exhibiting similar

15  dynamics are spatially clustered. However, I can envision cases in which the features defining the socio- hydrological dynamics are social not physical features. In these cases, I am not sure the 'spaces' would be contiguous. How would this approach be applied to a case where geographical features are poorly aligned with system dynamics? Or is this tool suitable for only the cases where geographical features are aligned with system dynamics?

2. In the definition section (pages 3-4), the authors present this concept/tool as an alternative to either narratives or

20  mathematical models. However, in the case that follows the authors present both the 'socio-hydrological space' delineation with a case narrative, which I think was effective. Rather than serving as an effective standalone tool, 'socio-hydrological spaces' compliments these other approaches. I think the author's argument for this tool would be more convincing if they could frame it as part of a broad research plan. For example, the authors note that SHS is descriptive not explanatory. If combined with other approaches could it enhance the explanatory power of a study?

25  3. While it is important to expand the approaches used to address socio-hydrological questions and to synthesize quantitative and qualitative data, this is not the first study to do so. The authors should acknowledge other efforts in this space such as data-driven narratives (Treuer et al. 2017) and the pairing of statistical analysis and narratives (Hornberger et al. 2015), and articulate what 'socio-hydrological spaces' adds.

4. I think there is potential for this concept to be used comparatively across say multiple flood plain cases. Please speak to

30  this potential.

5. Lastly, there are some typographic errors and awkward phrasing in the manuscript and it would benefit from a thorough review.

5 **Comments from Anonymous Referee #4**

The authors propose a new mid-level method for socio-hydrological research, which is a possible connection between the narratives of case studies and socio-hydrological models and allows for the detection of different socio-hydrological processes in real world cases studies and connect them with generic patterns described in socio-hydrological models.

The method is an interesting addition to socio-hydrological research and the manuscript very well describes the method and
10  its application. The introduction makes the case for SHS and defines where in socio-hydrological research SHS is placed and the analysis of the SHS Jamuna river floodplain is a good example of the application of this new method. However, I think the connection with the general processes described with socio-hydrological models is still a bit weak. I miss an extra step at the end of the paper where the authors describe the different socio-hydrological feedbacks that explain the behavior of each SHS (perhaps the authors could construct some causal loop diagrams based on the analysis they did). The conclusion now is
15  that SHS 1 is fighting, SHS 2 is adapting and SHS 3 is a mix of the two. This is very general and I think the authors could and perhaps should try to come up with some sub-patterns (i.e. there are different types of adaptation going on in these SHS) that they find in their analysis.

In section 6.2 the authors discuss the differences in flood frequency between the SHS and refer to figure 4b. It seems that figure 4b does not report anything about frequency, since it only shows the percentage of households that reported flooding?
20  Does the information about frequency come from other data that is not reported here? In that case I would change the figure. Otherwise the analysis in the text should be revised.

**2 Author's final response to editor's decision and guidance and final response to referees**

**General response to editor and reviewers**

We are grateful that the editor thinks there is a case for our paper to be published if we address the remaining issues. It is clear from the reviewers' comments that we did not yet explain clearly enough what our proposal entails = using socio-hydrological spaces as a new research approach that complements existing approaches. We revised especially Sections 1 and 3 to explain this better. We have implemented the following general changes in the paper:

- (particularly in response to reviewer #1) We emphasise that we are not looking to propose a way to model but to find patterns, which are two different research approaches to analysing data. The patterns can not only complement conceptual and/or deterministic modelling, but can also be used to understand and compare cases more qualitatively. Our approach is therefore adductive, i.e. to let the (qualitative and/or quantitative) data 'speak' for themselves while being guided by expectations from previous research, in this case the patterns 'fight' and 'adapt'. Our text has been carefully revised to be clearer on this point, especially Section 1.

- (particularly in response to reviewers #2 and #4) These reviewers seem to suggest that 'narrative' is a separate research method. They oppose narratives to statistics (reviewer #2) or models (reviewer #4). However, all research produces narratives, because they describe the findings (after all, scientific papers are mainly text). Relevant dichotomies to describe research methods are quantitative/qualitative, deductive/adductive, predictive/descriptive. We have been more explicit about the choices that we made in terms of these dichotomies.

We now respond to the specific comments made by the reviewers

**Reflection on other comments by the referees**

**Reply to anonymous referee #1**

We appreciate the constructive comments provided by Referee #1, which will help us improve the description of our work, though we are taken aback by some of her/his negative charges.

We agree with her/his suggestion that we could better "demonstrate how combining such rich historical patterns can help mutual reinforcing feedback between empirical studies and modeling studies". This is also requested by Referee #4. We have expanded the discussion on this topic. We emphasise that we are not looking to propose a way to model but to find patterns, which are two different research approaches. The patterns can not only complement conceptual and/or deterministic modelling, but can also be used to understand and compare cases more qualitatively. It is therefore a middle way between data and models, not between modelling in case studies and generic models. We do not propose "different style of modelling" but a different approach to research: to find patterns. Our previous text was not clear enough on this issue. Our text has been carefully revised to be clearer on this point, especially in Section 1 but changes have been made throughout the paper (highlighted by tracked changes). By implication, we now emphasise more that (historical) patterns are important.

However, we disagree that Mostert (2017) is an example to follow, since we propose a new research approach, not a combination of existing ones.

**Reply to anonymous referee #2**

5    We appreciate the constructive comments provided by Referee #2, which helped us improve the description of our work.

- We agree with the reviewer that SHS should be combined with other approaches than on its own. We clarify this point in the revised manuscript. This sentence at the end of Section emphasises this: "By providing locally relevant details and texture to more generically deduced patterns, SHS provides a useful methodological addition to the socio-hydrological understanding of floodplains." Also, our revised text for Section 1 is more explicit about pattern finding as complementary to modelling.

- As for policy relevance, the fact that use of SHS gives incentives to go to the field, talk to inhabitants and officials, and obtain a thorough understanding of the specifics of the location already means make socio-hydrological analyses potentially more policy-relevant than office-based modelling based on available data only. In terms of practical use of SHS, it can for instance be added as additional element to rapid rural appraisals, or other social assessments, to draw attention to how material conditions (hydrological and technical/infrastructure) co-shape social situations. We have expanded the text in this issue (Section 8).

- The reason why we changed the classification from levee effect/adaptation effect to fight/adapt is that we realised that we needed to separate the immediate response (fight or adapt) from the impacts this can have (levee effects and adaptation effects) and the different ways (technologies, behaviour) fighting or adapting is implemented. Fight/adapt are generic, while the resulting socio-hydrological changes are location specific. If the latter are taken into account, every SHS is different, as the reviewer says as well: "the multiple dimensions of human-water interactions can result in several distinct types of outcome patterns". This means that no truly generic patterns can be distinguished in 'levee effect' and adaptation effect'.

- We have also acknowledged that further study of the area reveals more details of the socio-hydrology of the spaces. However, the purpose of this paper as to introduce the concept SHS; further publications will expand on the details we found.

- We will include more details on the Chi-square test and ANOVA test in the supplemental information.

**Reply to anonymous referee #4**

30    We appreciate the constructive comments provided by Referee #4, which helped us improve the description of our work.

- Reviewer #4's concerns are mostly addressed by our actions on the comments by reviewer #1 and #2. We emphasise that we are not looking to propose a way to model but to find patterns, which are two different research approaches. The patterns can not only complement conceptual and/or deterministic modelling, but can also be used to understand and compare cases more qualitatively. It is therefore a middle way between data and models, not between modelling

in case studies and generic models. We do not propose "different style of modelling" but a different approach to research: to find patterns. Our previous text was not clear enough on this issue. Our text has been carefully revised to be clearer on this point, especially in Section 1 but changes have been made throughout the paper (highlighted by tracked changes).

- We also expanded on our conclusion that SHS 1 is fighting, SHS 2 is adapting and SHS 3 is a mix of the two. There are indeed sub-patterns within these spaces; these are explored in our further work together with the feedback loops that constitute the socio-hydrological character of each SHS. However, the purpose of this paper as to introduce the concept SHS; further publications will expand on the details we found.

- Regarding the use of 'narrative' as separate research method (and opposed to modelling): narrative is one qualitative research method amongst several, and can be used at case study level as well as for generic studies. Narratives can be combined with other qualitative data and also with quantitative data, as we do in our paper. We have edited the text to be more explicit about these dichotomies in our paper: quantitative/qualitative research methods, and case study vs generic study.

- We thank the reviewer for pointing out the mistake in Figure 4b. The figure title was indeed incorrect and will be changed to 'flood occurrence'.

**3 Author's changes in manuscript (Second Revision)**

**Socio-hydrological spaces in the Jamuna River floodplain in Bangladesh**

[revised manuscript text omitted]

prove that these boundaries to the SHS are the best, since this would require much more detailed social and hydrological data
than currently available. However, we did show~~

[revised manuscript text omitted]

---

## Author Response (AR3)

**Socio-hydrological spaces in the Jamuna River floodplain in Bangladesh**

Md Ruknul Ferdous, Anna Wesselink, Luigia Brandimarte, Kymo Slager, Margreet Zwarteveen, and Giuliano Di

Baldassarre

**Response Letter**

5

**1 Comments from editor and referee**

**Editor's comments and guidance:**

**Editor Decision: Minor revisions**

Dear Authors, I now have three reviews of your revised manuscript, two from the previous set of reviewers and one entirely

10 new one. The previous authors are split in their decisions, one recommending acceptance and the other still not satisfied with the revisions, and is still convinced about the SHS concept. The third, new reviewer does like the concept and recommends acceptance.

I have decided to accept the paper subject to minor/moderate revisions you can make to accommodate the misgivings of the second reviewer in a way that this can be a subject of further conversations and investigations.

15 It will NOT be sent for further external review, and I will quickly review and if I am satisfied I will forward it for publication in HESS. I am sorry that it has taken so long, but hope you agree that the review process has resulted in a much improved manuscript.

I look forward to receiving your revised, final version.

**Author's response to editor's decision and guidance:**

20 We would like to thank again the Editor, Prof. Sivapalan, for handling the revision process of our paper. We also acknowledge all the referees (including the ones of the previous versions) for providing very useful and constructive comments that have helped us improve the description of our work. Indeed, we do agree that you agree that the review process has resulted in a much improved manuscript.

We are glad to see that the new referee likes our concept and recommends acceptance, and we are also happy to hear that we

25 convinced with our revision work one more reviewer.

We understand that Anonymous Referee #4 remains unconvinced. As suggested by the Editor, we further revised our manuscript to accommodate her/his misgivings. In particular, we addressed her/his specific questions. We believe that this effort also helped clarify more the concept of socio-hydrological space (SHS), and its middle ground place between case studies and generic models in socio-hydrology.

**Comments from anonymous referee #4**

The authors present a new method for the study of socio-hydrological systems. They claim their method provides a way to detect patterns in cases studies that can be connected to general patterns that have been described in the literature. While a research method that covers this kind of connections may be useful, I am still not convinced by the way it is presented in this

- 5 paper. To me it seems that the description and analysis of the case study in this paper is not very different from other case study descriptions. The analysis connecting the different hydrological and social patterns could be more detailed and the patterns of adapt and fight that the authors detect seem to still be very general. If the method is a way to fill the gap between patterns found in case studies and patterns described with generic models I would expect the analysis of the patterns in this case study to be a bit more detailed.
- 10 In the introduction the authors state that: "As in any attempt to produce insights that transcend specific cases, methods of abstraction from reality to find causal relationships and stylised equations 15 (generalisation) are sometimes difficult to reconcile with more detailed representations of what is happening in a specific location (Blair & Buytaert, 2016). While enabling global comparison by using data sets from different locations, generic models unavoidably foreground some elements or dimensions of flood-society dynamics to the neglect of others (Magliocca et al., 2018)." Is your representation
- 15 of these three spaces as either fight or adapt or a combination not just as simple as the ones described with generic models?

**Author's response to anonymous referee #4:**

We thank the Anonymous Referee #4 for providing criticism to our paper, which has helped us strengthen the description of our concept and its application to a floodplain in Bangladesh. As stated in the revised paper, "pattern detection is no new activity in socio-hydrology, because patterns are at the basis of the stylised representations (equations) in generic models.

- 20 Historical patterns are foundational for a full understanding of generic as well as place-based models, and pattern finding reinforces the feedback between empirical studies and modelling studies. However, we propose a new socio-hydrological concept to operationalise the search for patterns in the messy reality of specific cases: socio-hydrological spaces (SHS). Eventually, patterns found in cases may be formalised into causal relationships, but this does not necessary have to be the goal." As such, our representation of these three SHSs as either fight or adapt or a combination are not just as simple as the
- 25 ones described with generic models. In fact, this socio-hydrological spaces are more bottom-up and allows for unexpected outcomes (see next point), while models are more top-down and might miss some elements. The revised manuscript states that "using SHS in the analysis of socio-hydrological dynamics helps to make the necessary intermediary step between the messy and many details used to characterize a specific location (space) and the stylised abstraction of generic models. With the proposal of SHS we are looking for a middle ground where we preserve the variability of reality and the unpredictability
- 30 of human behaviour and decisions, not force-fitting these into a model, while at the same time recognising patterns (due to combinations of similar or comparable fight and/or adapt responses). We thus propose that SHS can serve the function of a lens through which to view and filter the complex reality of specific cases, in order to find patterns in human-water interactions. Such patterns can then be compared and contrasted to patterns in other locations to see if further generalisation

towards generic models is possible." More interestingly, the use of SHS invites the researcher to have an open mind to the existence of expected or **unexpected** patterns in the location under investigation, using a thorough understanding of the specifics of this location in terms of society, history, economics, natural system, technical interventions, etc. ". The use of bold in "unexpected" is intended. See below.

5

**Comments from anonymous referee #4**

In the abstract the authors present their study as: "Our example of the use of SHS shows that the concept draws attention to how historical patterns in the co-evolution of social behaviour, natural processes and technological interventions give rise to different landscapes, different styles of living, and different ways of organizing livelihoods." I do not really find these

10 patterns of co-evolution in the descriptions of the case study. Instead to me it is a description of the separate social, economic, hydrologic, etc findings, and I miss an attempt to combine these to determine what different co-evolution patterns can be found in this particular case study.

**Author's response to anonymous referee #4:**

The paper does links "social, economic, hydrologic, etc... findings" everywhere throughout the manuscript. Indeed, it does not attempt to make an "explanation by feedback mechanisms", which is probably what the Referee is looking for, but that's

15 not attempt to make an "explanation by feedback mechanisms", which is probably what the Referee is looking for, but that's the main goal of socio-hydrological modelling. We made these interlinks more explicit in the revised manuscript, by discussing the interactions between the different panel of Figure 4. See below.

The Referee raises specific questions below that we are happy to respond and that we believe helped us sharpen our message and clarify more the potential of socio-hydrological space (SHS) in finding **unexpected** patterns.

20

**Comments from anonymous referee #4**

For example, the people in SHS2 have a higher income than in SHS1 and they seem to experience floods the least frequent, but still they have the highest amount of houses made of earthen floor, wood and bamboo mats. Why is that? Is it because there is a different pattern of fight and adaptation in this SHS than in the others?

**25 Author's response to anonymous referee #4:**

This is not correct. In fact, Figure 4 shows the opposite. People in SHS2 have a lower (not higher) income than in SHS1 and they experience floods more (not less) frequently.

**Comments from anonymous referee #4**

30 Also the damages in SHS2 seem to be lower than in the other SHS (with the exception of the flood in 1988), is this because the people there are already poorer and thus there is less to damage? Or because they are better adapted and are able to reduce the damages because of flooding?

**Author's response to anonymous referee #4:**

This is an example of unexpected pattern that could not been foreseen by current models of human-flood interactions. Flood damage in SHS2 are lower for both reasons: i) people there are already poorer, ii) people there are better adapted (as they get flooded every single year, see Figure 4b). But, while they have adapted to frequent flood events, this does not make them

5 less vulnerable to big floods, such as the one of 1988 (see Figure 4c). This outcome is unexpected, as it would not be captured by any of the current socio-hydrological models of human-flood interactions. We have highlighted this point in the revised manuscript (page 15, lines 6-11):

"It is interesting to observe that (apart from the 1988 event), flood damage in SHS2 is lower than damage in SHS1 and SHS3. This is not only because people there are generally poorer (Fig. 4e), but also because people there are better adapted

10 to frequent flooding (as they get flooded every year, see Fig. 4b). Yet, while people in SHS2 have adapted to frequent flood events, this adaptation does not make them less vulnerable to big floods, such as the one of 1988 (see Figure 4). This outcome was unexpected, and it would not be captured by any of the current models of human-flood interactions proposed so far."

**15 Comments from anonymous referee #4**

People in SHS3 have more experience with migration than in SHS1, but they flood less frequently, is there some explanation for that? These are the kind of connections I would expect from an analysis of "how historical patterns in the co-evolution of social behaviour, natural processes and technological interventions give rise to different landscapes, different styles of living, and different ways of organizing livelihoods."

**20 Author's response to anonymous referee #4:**

We agree with the Referee that this are the interesting connections to explore and study in socio-hydrology. The manuscript already stated that this was due to the fact the riverbank erosion drives migration more than flooding, but we have made this point more clear in the revised manuscript (page 17 lines 7-8) by linking together more explicitly different panels of figure 4: "The study shows that riverbank erosion (Fig. 4d), more than flooding (Fig. 4b), is one of the main drivers for relocation from their place of origin (Fig. 4f)."

**Comments from anonymous referee #4**

25

30

I would suggest to the authors to either rephrase their approach or application of SHS in a way that it indeed adds something new to the analysis of a case study (which I am still not convinced it does now) or to drop the concept of SHS and focus on the case study itself.

**Author's response to anonymous referee #4:**

We thank again the Anonymous Referee #4 for providing useful criticism that we believe has helped us improve substantially the description of the SHS concept and its complementary relation with case studies and generic models in socio-hydrology.

[revised manuscript text omitted]